# TiViT: Time Series Representations for Classification Lie Hidden in Pretrained Vision Transformers

**Simon Roschmann** [1 2 3 4]  **Quentin Bouniot** [1 2 3 4 5]  **Vasilii Feofanov** [6 7]  **Ievgen Redko** [6]  **Zeynep Akata** [1 2 3 4]

## Abstract

Adapting vision models for time series analysis is compelling, yet all existing approaches are falling short of dedicated time series foundation models (TSFMs) in classification. We propose **Ti**me **Vi**sion **T**ransformer (**TiViT**), the first framework to successfully unlock the representational power of frozen Vision Transformers (ViTs) pretrained on large-scale image datasets for time series classification. By using hidden representations of OpenCLIP models, TiViT achieves state-of-the-art performance on time series classification benchmarks without finetuning. We analyze the representations of TiViT and find that intermediate ViT layers with high intrinsic dimension are the most effective. We further assess the alignment between TiViT and TSFM representation spaces, revealing strong complementarity and additional gains through feature concatenation. Finally, we unfreeze the ViT backbone for continual pretraining on synthetic time series. Code is available at https://github.com/ExplainableML/TiViT.

## 1. Introduction

Inspired by the success of foundation models in natural language processing (NLP) and computer vision (CV), similar models have been developed for the analysis of time series following two approaches. The first one is to pretrain time series foundation models (TSFMs) in a self-supervised way (Ansari et al., 2024; Das et al., 2024; Feofanov et al., 2025; Goswami et al., 2024; Lin et al., 2023) using a large-scale real-world time series dataset. The second one is to repurpose powerful foundation models from other domains, such as NLP (Jin et al., 2024; Zhou et al., 2023) and CV (Chen et al., 2024; Li et al., 2023b), for time series tasks. The idea behind these approaches is to benefit from the vast amount of samples that large vision and language models are trained on and which are unavailable in the time series domain.

Time series can be transformed into images in various ways, e.g., by rendering the signal in the time domain (line plot) or frequency domain (spectrogram), or by using a 2D modeling approach (heatmap, Gramian angular field, recurrence plot) that stacks segments of the signal based on a chosen periodicity. Vision models, often based on convolutional neural networks and their variants, have been applied to these representations, but most were supervised models trained for individual datasets (Ni et al., 2025). This work explores how pretrained vision foundation models such as OpenCLIP (Radford et al., 2021; Ilharco et al., 2021), SigLIP 2 (Tschannen et al., 2025), and DINOv3 (Siméoni et al., 2025) can be used as powerful feature extractors without pretraining or finetuning on time series data. Li et al. (2023b) showed that pretrained ViTs can be effective in the classification of irregular time series from their line plot representations after full finetuning. In a similar vein, Chen et al. (2024) applied a masked auto-encoder with a pretrained frozen ViT to 2D transformed time series to perform time series forecasting. Different from these works, our TiViT model surpasses the performance of frontier TSFMs across a broad set of classification benchmarks. Moreover, we show that pretrained ViTs serve as a powerful initialization for time series encoders and can be strengthened through continual pretraining on time series. While prior work has considered 1D and 2D modeling approaches separately, we are the first to identify and leverage their complementarity.

Our contributions are threefold: (1) We introduce the Time Vision Transformer (TiViT), which leverages hidden representations of pretrained frozen ViTs for time series classification. TiViT surpasses conventional TSFMs without finetuning across 128 classification datasets. (2) We study the alignment between TiViT and TSFMs and find that they extract complementary information. Merging their representations improves average classification performance by $+3\%$ over TSFMs. (3) We unfreeze the TiViT backbone for continual pretraining on synthetic time series, enabling a lightweight TiViT variant with 35M parameters to close the gap to state-of-the-art classification performance.

[1]Helmholtz Munich [2]Technical University of Munich [3]Munich Center for Machine Learning [4]Munich Data Science Institute [5]Télécom Paris [6]Huawei Noah's Ark Lab [7]42.com. Correspondence to: Simon Roschmann <simon.roschmann@tum.de>.

*Proceedings of the $2^{nd}$ ICML Workshop on Foundation Models for Structured Data*, Seoul, South Korea. 2026. Copyright 2026 by the author(s).

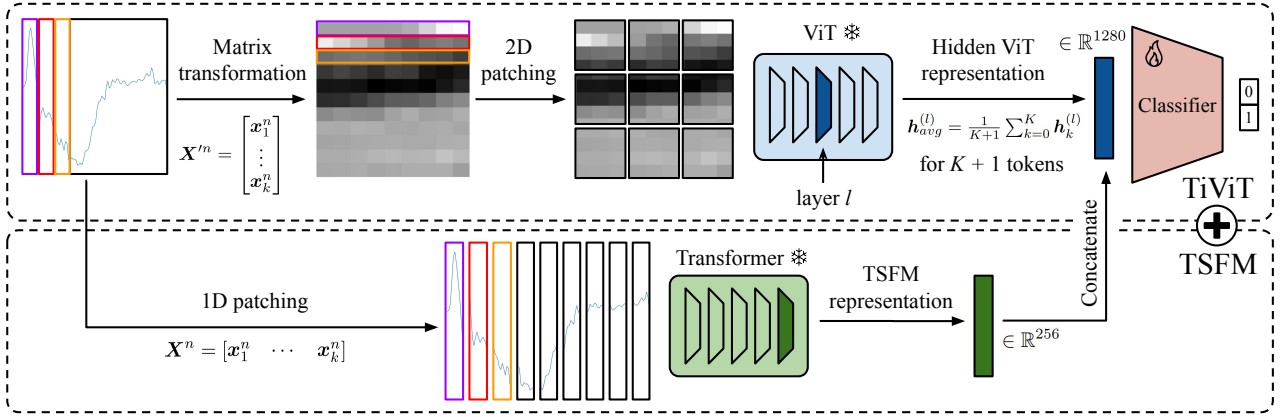

*Figure 1.* Illustration of TiViT on a time series sample from ECG200 (Olszewski, 2001). We split the time series into segments and stack them to form a grayscale image. Then, we patch the image in 2D and feed it into a frozen ViT pretrained on large-scale image datasets. We average the representations from a specific hidden layer and pass them to a learnable classification head. Concatenating the representations of TiViT and TSFMs prior to classification further improves accuracy.

## 2. Time Vision Transformer

We introduce the Time Vision Transformer (TiViT) leveraging pretrained frozen ViTs from the vision or vision-language domain for time series classification. We consider a multivariate time series dataset $\mathcal{T} = \{ \boldsymbol{t}^n | \boldsymbol{t}^n \in \mathbb{R}^{T \times D} \}_{n=1}^N$ containing $N$ samples, each of length $T$ and dimensionality $D$. The corresponding targets $\mathcal{Y} = \{ y^n \}_{n=1}^N$ are labels $y^n \in \{ 1, ..., C \}$ from $C$ different classes. We transform the time series into images and apply ViTs on these images to extract representations for linear classification. Figure 1 illustrates our approach.

### 2.1. Time Series-to-Image Transformation

Following the channel independence assumption proposed by Nie et al. (2023), we first split a multivariate time series $\boldsymbol{t}^n \in \mathbb{R}^{T \times D}$ into $D$ univariate time series $\{ \boldsymbol{t}_d^n \in \mathbb{R}^T \}_{d=1}^D$. We then normalize each univariate time series $\boldsymbol{t}_d^n$ using robust scaling, defined as: $\frac{\boldsymbol{t}_d^n - Q_2}{Q_3 - Q_1}$, where $Q_1, Q_2, Q_3$ are the first, second (median), and third quartiles, respectively. We apply padding at the beginning of each time series by replicating its first value and subsequently segment it into $M$ patches $\{ \boldsymbol{x}_m \}_{m=1}^M$ of size $P$. Given a patch length $P$ and stride $S$, the total number of patches is: $M = \lfloor \frac{T-P}{S} \rfloor + 1$. We stack the patches to generate a 2D representation $\boldsymbol{X}' \in \mathbb{R}^{M \times P}$, which we then render into a grayscale image $\boldsymbol{X}' \in \mathbb{R}^{M \times P \times 3}$ by replicating its signals across three channels. To align with the square input resolution $(R, R)$ expected by the ViT, we resize the image.

### 2.2. Time Series Classification with Frozen ViTs

We feed each grayscale image $\boldsymbol{X}'$ representing a univariate time series into a pretrained and frozen ViT $v$ with $L$ hidden layers. The ViT inherent 2D patching yields a sequence $\{ \boldsymbol{x}_k' \in \mathbb{R}^{U^2} \}_{k=1}^K$ of flattened patches where $(U, U)$ is the resolution per patch and $K = R^2/U^2$ is the resulting number of patches. ViTs generally prepend a classification token to this sequence. The ViT consumes all input tokens and produces a sequence of features at every layer: $v(\boldsymbol{X}') = \left\{ [\boldsymbol{h}_0^{(l)}, \boldsymbol{h}_1^{(l)}, ..., \boldsymbol{h}_K^{(l)}] \right\}_{l=0}^L$. To obtain a single embedding vector $\boldsymbol{e}$ per image, we select a specific layer $l$ and average its $K + 1$ representations: $\boldsymbol{e} = \boldsymbol{h}_{avg}^{(l)} = \frac{1}{K+1} \sum_{k=0}^K \boldsymbol{h}_k^{(l)}$. For multivariate time series, we feed per-channel image representations $\{ \boldsymbol{X}_d' \}_{d=1}^D$ separately into the ViT and concatenate the resulting embeddings for a specified layer: $\text{Concat}(\boldsymbol{e}_1, ..., \boldsymbol{e}_D)$. We only train a linear classifier on the ViT representations and their corresponding class labels.

### 2.3. Continual Pretraining on Time Series

We further adapt TiViT to time series using synthetic data from CauKer (Xie et al., 2026). Similar to Mantis (Feofanov et al., 2025), we use contrastive pretraining: two augmentations of the same time series are pulled together, while different samples should be well separated. The augmentations are based on RandomCropResize with crop rate $c \in [0, 0.2]$. We truncate the ViT at the best layer $l$ selected by linear probing and add a lightweight projection head. Given two augmented views, we transform them into images, encode them with TiViT, and optimize a symmetric InfoNCE loss (Oord et al., 2018): $\mathcal{L} = \frac{1}{2} \left( \text{CE}(S/\tau) + \text{CE}(S^\top/\tau) \right)$, where $S$ contains pairwise cosine similarities and $\tau$ is a learnable temperature initialized to $0.1$. During pretraining, we update only the projection head, LoRA adapters, and layer normalization parameters. Afterward, we discard the projection head and use the aggregated backbone representation $\boldsymbol{h}_{avg}^{(l)}$ for downstream tasks.

*Table 1.* Classification accuracy of TSFMs and frozen TiViT.

| Model | UCR | UEA | WOODS |
|---|---|---|---|
| Moment | 79.0 | 69.9 | 70.4 |
| Mantis | 80.1 | 72.4 | 70.1 |
| TiViT *(Ours)* | 81.6 | 72.0 | 72.4 |
| TiViT + Moment | 82.7 | 72.6 | 73.5 |
| TiViT + Mantis | **83.1** | **73.7** | **73.7** |

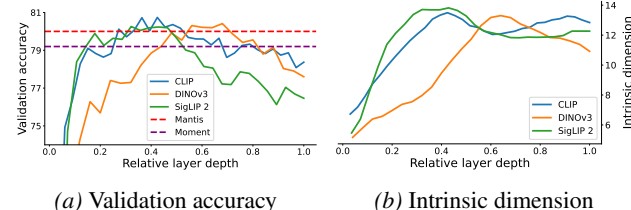

*(a)* Validation accuracy  *(b)* Intrinsic dimension

*Figure 3.* Accuracy and intrinsic dimension with hidden representations at different depths of pretrained ViTs. Results are averaged over 128 datasets from the UCR benchmark.

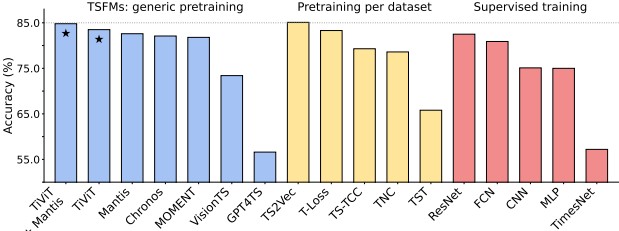

*Figure 2.* Classification accuracy on 91 UCR datasets.

## 3. Experimental Evaluation

We evaluate TiViT with three different ViT backbones (CLIP, SigLIP 2, DINOv3) on the UCR (Dau et al., 2019), UEA (Bagnall et al., 2018), and WOODS (Gagnon-Audet et al., 2023) benchmarks for time series classification. We compare TiViT to the state-of-the-art TSFMs Mantis (Feofanov et al., 2025) and Moment (Goswami et al., 2024) pretrained on time series. We further consider GPT4TS (Zhou et al., 2023) pretrained on textual data, vision-based approaches (VisionTS (Chen et al., 2024), TimesNet (Wu et al., 2023)), forecasting TSFMs (VisionTS (Chen et al., 2024), Chronos (Ansari et al., 2024)), and a wide range of (self-)supervised baselines. Our experimental setup is detailed in Appendix A.

### 3.1. Transforming Time Series into Images

The performance of our time series-to-image transformation is sensitive to the time series patch size $P$, as extreme values can create redundant visual tokens during resizing to the ViT input resolution. To avoid a computationally expensive hyperparameter search for the optimal patch size $P^*$ per dataset, we propose the heuristic $P = \sqrt{T}$ for any series of length $T$. While an exhaustive search for $P^*$ offers a marginal accuracy improvement in the case of no overlap, our heuristic provides a strong baseline at a fraction of the computational cost. As displayed in Appendix B.1 (Figure 6), introducing overlap between patches further boosts performance and makes the impact of the optimal patch size vanish. Consequently, we use a patch size of $P = \sqrt{T}$ and a stride of $S = P/10$ in the following experiments.

## 3.2. Time Series Classification with Frozen ViTs

We apply frozen ViTs as feature extractors on images generated from time series. Here, we refer to our best performing model with a ViT-H backbone from OpenCLIP (up to layer 14) as TiViT. Table 1 displays the comparison of TiViT and TSFMs on the UCR and UEA test set. The state-of-the-art TSFM Mantis achieves a classification accuracy of 80.1% on the UCR benchmark. Our statistical analysis with a paired t-test and a significance level of 0.05 confirms that TiViT significantly outperforms ($p = 0.003$) Mantis across the 128 datasets of the UCR benchmark, achieving 81.6% accuracy. We further extend our analysis to multivariate time series. TiViT reaches a classification accuracy of 72.0%, which is statistically on par with Mantis on the UEA benchmark.

Figure 2 shows that TiViT outperforms not only other TSFMs, but also a series of (self-)supervised baselines from Goswami et al. (2023) (pre-)trained per dataset. VisionTS has repurposed pretrained masked autoencoders (MAEs) for time series forecasting. However, when we apply VisionTS in time series classification on the UCR datasets, it achieves only 73.4% accuracy and remains behind TiViT with an accuracy of 83.5%. This performance gap cannot be explained by the choice of the vision backbone alone, but by a crucial modeling insight. We find that the most effective features for classification lie in the hidden layers rather than the output layers of pretrained ViTs (see Section 3.3). TimesNet is another vision approach trained in a supervised manner on each time series dataset. It is computationally expensive and reaches only 57.2% accuracy. Interestingly, TSFMs such as Chronos and VisionTS, primarily designed for forecasting, perform worse than TiViT or Mantis in classification. This highlights that models optimized for forecasting cannot be simply transferred to classification tasks and emphasizes the need for classification-focused TSFMs like TiViT.

### 3.3. Effectiveness of Hidden ViT Representations

While the final representations of ViTs typically capture high level semantics, intermediate layers encode lower level information (Dorszewski et al., 2025). Our study reveals that the intermediate representations of ViTs are the most effective for time series classification. Figure 3a reports

*Table 2.* Joint classification accuracy and alignment score (mutual-kNN) for TiViTs and TSFMs on the UCR benchmark.

| Fusion | Model 1 | | Model 2 | | Joint accuracy | Alignment score |
|---|---|---|---|---|---|---|
| | Name | Acc | Name | Acc | | |
| TSFM + TSFM | Mantis | 80.1 | Moment | 79.0 | 81.5 | 0.222 |
| TiViT + TiViT | CLIP | 81.6 | DINOv3 | 80.2 | 82.2 | **0.431** |
| TiViT + TSFM | DINOv3 | 80.0 | Moment | 79.0 | 82.0 | 0.213 |
| | DINOv3 | 80.0 | Mantis | 80.1 | 82.5 | 0.243 |
| | CLIP | 81.6 | Moment | 79.0 | 82.7 | 0.241 |
| | CLIP | 81.6 | Mantis | 80.1 | **83.1** | 0.262 |

*Table 3.* Classification accuracy of TiViT on the UCR benchmark with ViT-B backbones (6 layers) after continual pretraining.

| TiViT | Frozen | Continual pretraining |
|---|---|---|
| DINOv3 | 80.2 | 81.2 +1.0 |
| SigLIP 2 | 79.1 | 80.5 +1.4 |
| CLIP | 80.8 | 81.8 +1.0 |

the accuracy of TiViT with pretrained ViTs from DINOv3, OpenCLIP, and SigLIP 2 on the validation split of the UCR benchmark. For each dataset, we extract representations from the hidden layers of ViTs, average them, and train a linear classifier. The intermediate representations of ViTs, between 40% and 70% of the layer depth, achieve the highest accuracy. Note that simply using the final layer like previous vision-based approaches would be far from state-of-the-art classification performance.

To better understand the hidden representations of ViTs, we analyze their intrinsic dimension (ID) on UCR datasets using the DADApy (Glielmo et al., 2022) implementation of the TWO-NN estimator (Facco et al., 2017). Figure 3b displays for different ViT backbones the intrinsic dimensionality of their representations at varying layer depth. The mean Pearson correlation coefficient between the intrinsic dimension and validation accuracy is $\rho = 0.704$. The best performing layers exhibit the highest or second highest ID.

### 3.4. Representational Alignment of TiViT and TSFMs

We study whether TiViT and TSFMs learn complementary representations and concatenate their features for joint classification. As shown in Table 2, combining two TSFMs improves accuracy to 81.5%, while combining TiViT-CLIP with Moment or Mantis reaches 82.7% and 83.1%, respectively. This indicates that TiViT captures information that is not yet contained in existing TSFMs. To better understand this complementarity, we measure representational alignment using mutual k-nearest neighbors (Huh et al., 2024) on the 10 largest UCR datasets. Alignment between TiViTs and TSFMs is low, suggesting that they process the same time series differently. This likely explains why their concatenation improves classification.

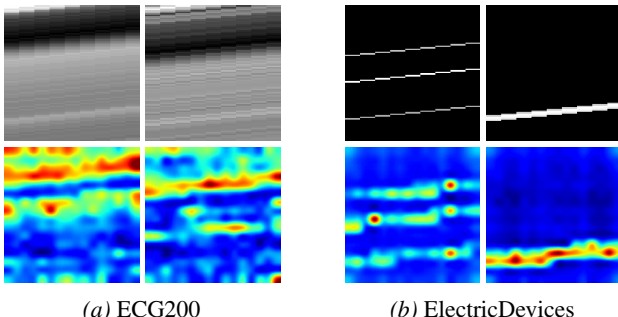

*(a)* ECG200        *(b)* ElectricDevices

*Figure 4.* Attention rollout.

### 3.5. Continual Pretraining

We adapt the ViT backbone of TiViT, pretrained on natural images, to the time series domain, via a second stage of continual pretraining on 2M synthetic time series from CauKer (Xie et al., 2026). Here, we focus on the smallest TiViT model with ViT-B backbone and inject LoRA adapters to make contrastive pretraining on 4 GPUs with an effective batch size of 4096 feasible. Details are provided in Appendix A. Table 3 shows the classification performance of TiViT on the UCR benchmark with various ViT-B backbones before and after continual pretraining. The contrastive learning on time series improves the average accuracy of all backbones across the 128 UCR datasets by more than 1%. Notably, TiViT with ViT-B backbone from OpenCLIP (36 M parameters, 81.8% accuracy) now outperforms the best frozen TiViT model with ViT-H backbone (257 M parameters, 81.6% accuracy) using $7\times$ fewer parameters.

### 3.6. Qualitative Feature Analysis

We apply attention rollout (Abnar & Zuidema, 2020) to TiViT on image representations from ECG200 and ElectricDevices. As shown in Figure 4, the aggregated attention highlights salient bright and dark regions, corresponding to high and low signals in the original time series. This suggests that TiViT focuses on discriminative temporal patterns. We further provide t-SNE visualizations of TiViT representations in Appendix B.11.

## 4. Conclusion

In this paper, we introduced TiViT, the first method to successfully leverage large pretrained ViTs for time series classification. TiViT with a frozen ViT backbone and hidden representations significantly outperformed TSFMs in time series classification on the UCR benchmark and reached competitive results on UEA. We highlighted the complementarity of TiViT and TSFMs, and by combining their representations, established the new state-of-the-art in time series classification.

## Acknowledgments

This work was partially funded by the ERC (853489 - DEXIM) and the Alfried Krupp von Bohlen und Halbach Foundation, which we thank for their generous support. The authors gratefully acknowledge the scientific support and resources of the AI service infrastructure *LRZ AI Systems* provided by the Leibniz Supercomputing Centre (LRZ) of the Bavarian Academy of Sciences and Humanities (BAdW), funded by Bayerisches Staatsministerium für Wissenschaft und Kunst (StMWK). This work was also supported by Hi! PARIS, and the ANR/France 2030 program (ANR-23-IACL-0005).

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

In Section A, we detail our experimental setup for classification, continual pretraining and alignment, as well as for anomaly detection and forecasting. In Section B, we analyze design choices of TiViT including the time series-to-image transformation or the size and type of the ViT backbone. In Section C, we provide the full benchmark results in classification, anomaly detection, and forecasting per dataset. In Section D, we present a theoretical insight and experiments showing the benefits of 2D patching over conventional 1D patching for time series. Finally, we discuss the broader impacts of our work in Section E.

## A. Experimental Setup

**Datasets** UCR (Dau et al., 2019) comprises 128 univariate time series datasets of varying sample size ($16 \leq N_{\text{train}} \leq 8926$) and series length ($15 \leq T \leq 2844$). UEA (Bagnall et al., 2018) consists of 30 multivariate time series datasets. Following Feofanov et al. (2025), we exclude three datasets (AtrialFibrillation, StandWalkJump, PenDigits) from UEA in our main evaluation due to their short sequence length or small test size. Furthermore, we consider 3 multivariate EEG datasets (PCL, CAP, SEDFx) from the WOODS (Gagnon-Audet et al., 2023) repository. These datasets come in different splits reflecting different research groups (PCL), machines (CAP), or age groups (SEDFx). For in-domain (ID) evaluation, we merge all splits before performing the train-val-test split. For out-of-domain (OOD) evaluation, we hold out one split for validation and another one for testing, while using the remaining splits for training.

**Vision Transformers** Our study mainly examines three differently pretrained ViTs: OpenCLIP (Cherti et al., 2023; Ilharco et al., 2021), SigLIP 2 (Tschannen et al., 2025), and DINOv3 (Siméoni et al., 2025). CLIP (Radford et al., 2021) performs contrastive learning of image and text encoders on image-text pairs. We reuse the ViT image encoders of OpenCLIP (Cherti et al., 2023; Ilharco et al., 2021) models trained with the LAION-2B English subset of LAION-5B (Schuhmann et al., 2022). SigLIP 2 (Tschannen et al., 2025) adopts contrastive learning on image-text pairs, but with a Sigmoid loss, complemented by captioning-based pretraining, self-distillation, and masked prediction. In contrast, DINOv3 (Siméoni et al., 2025) is solely pretrained on images through self-distillation with a student-teacher architecture and objectives at both the image and patch level. For each pretraining approach, we consider multiple vision model sizes (ViT-B, ViT-L, ViT-H) with varying layer depth (12, 24, and 32 layers). Additionally, we investigate the effectiveness of ViTs from DINOv2 (Oquab et al., 2024) and Masked Autoencoders (He et al., 2022) in the appendix.

**Baselines** We compare TiViT to two state-of-the-art TSFMs exclusively pretrained on time series. Mantis (Feofanov et al., 2025) is a Transformer model (8 M parameters) comprising 6 layers and 8 heads per layer, pretrained on 2 million time series with contrastive learning. As stated by Feofanov et al. (2025), Mantis is based on the ViT architecture, making it particularly suitable for our comparison with large-scale ViTs trained on natural images. Moment (Goswami et al., 2024) is a family of Transformers pretrained on 13 million time series with masked modeling. In our study, we consider Moment-base with 12 layers and 125 M parameters.

We further consider GPT4TS (Zhou et al., 2023) pretrained on textual data and a wide range of supervised and self-supervised baselines (pre-)trained per time series dataset. The 9 supervised baselines comprise: ResNet (Wang et al., 2017), FCN (Wang et al., 2017), DTW (Dau et al., 2019), CNN (Zebik et al., 2017), MLP (Wang et al., 2017), Encoder (Serrà et al., 2018), TWIESN (Tanisaro & Heidemann, 2016), MCNN (Cui et al., 2016), and TimesNet (Wu et al., 2023). The 5 self-supervised baselines are: TS2Vec (Yue et al., 2022), T-Loss (Franceschi et al., 2019), TS-TCC (Eldele et al., 2021), TNC (Tonekaboni et al., 2021), and TST (Zerveas et al., 2021). For all of these baselines, we utilize the classification accuracy reported by Goswami et al. (2024) in our comparison.

Furthermore, we evaluate the effectiveness of two state-of-the-art TSFMs that have been designed for time series forecasting in time series classification: Chronos Bolt Base (Ansari et al., 2024) and VisionTS (Chen et al., 2024) with MAE Base backbone. We average the sequence of their output representations to obtain a single representation for linear classification.

**Classification** To assess the effectiveness of TiViT and TSFM representations in time series classification, we train a logistic regressor with the LBFGS solver per dataset. Our evaluation adheres to the standard train-test splits provided by the UCR and UEA archive and reserves 20% of the train split for validation. For the time series-to-image transformation, we resize the grayscale images to the resolution expected by the ViT with nearest interpolation and adjust the contrast with a factor of 0.8. To compute the mutual kNN alignment score between models, we select the 10 largest UCR datasets, sample 1024 time series from each dataset, and measure the overlap of their representations for k=10. This setup is in line with Huh et al. (2024). All linear probing experiments can be performed on a single NVIDIA V100 GPU with 16 GB memory.

**Continual pretraining and contrastive alignment** To bridge the gap between the image and time series domain, we further adapt the ViT-B backbones through continual pretraining or alignment with TSFMs on time series data. We focus on TiViT

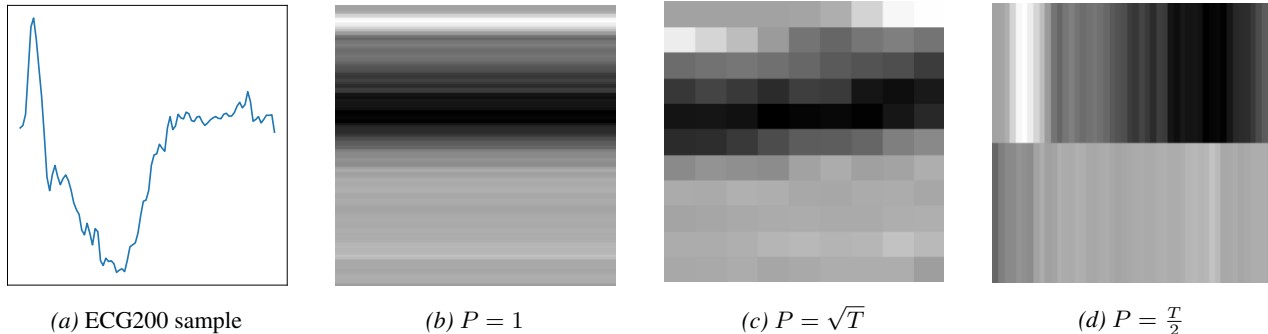

*(a) ECG200 sample*  *(b) $P = 1$*  *(c) $P = \sqrt{T}$*  *(d) $P = \frac{T}{2}$*

*Figure 5.* Effect of patch size $P$ on the time series-to-image transformation of a sample from the ECG 200 (Olszewski, 2001) dataset. To match the ViT input resolution, a small patch size ($P = 1$) requires horizontal stretching, while a large patch size ($P = \frac{T}{2}$) requires vertical stretching. Both scenarios result in redundant tokens.

*Table 4.* Effect of patch size and overlap on UCR classification accuracy.

| Patch size | $\sqrt{T}$ | | | $P^*$ | | |
|---|---|---|---|---|---|---|
| Overlap | 0.0 | 0.5 | 0.9 | 0.0 | 0.5 | 0.9 |
| Val accuracy | 78.0 | 80.3 | 80.7 | 88.1 | 88.9 | 89.7 |
| Test accuracy | 78.3 | 80.4 | 81.6 | 79.3 | 80.7 | 81.7 |

with a ViT-B backbone to make this adaptation with a sufficiently large batch size on 4 NVIDIA H100 GPUs feasible. We retain the backbone up to the layer that achieves the best linear probing performance. For all ViT-B backbones from OpenCLIP, SigLIP 2, and DINOv3, this corresponds to layer 6. On top of the truncated backbone, we append lightweight projection modules. For continual pretraining, we add a single Transformer layer with a learnable CLS token, followed by an MLP with one hidden layer that projects the CLS token into a joint embedding space. For contrastive alignment, we use only an MLP with one hidden layer on top of the aggregated representations of TiViT and the TSFM.

We train all models on 2M synthetic time series from CauKer (Xie et al., 2026) for up to 5 epochs using LoRA (Hu et al., 2022) with rank $r = 8$, scaling factor $\alpha = 16$, and dropout 0.1. LoRA is applied primarily to the query and value projections; when these are not separately available, it is applied to the output projection. We additionally adapt the layer normalization parameters. Training uses a cosine annealing learning rate scheduler with a maximum learning rate of $10^{-5}$, a warmup ratio of 0.05, and a batch size of 4096. For continual pretraining, we apply RandomCropResize as augmentation with a cropping rate $c \in [0, 0.2]$. After pretraining or alignment, we discard the projection heads and utilize the adapted backbone in downstream classification.

**Anomaly detection** For this task, we equip TiViT with 6 layers of OpenCLIP ViT-B, apply no patch overlap, and flatten the sequence of representations before learning a linear reconstruction head per dataset. TiViT is evaluated across 248 datasets from the UCR Anomaly Archive (Wu & Keogh, 2023) and compared to the following baselines: Moment (Goswami et al., 2024), GPT4TS (Zhou et al., 2023), TimesNet (Wu et al., 2023), Anomaly Transformer (Xu et al., 2022), DGHL (Challu et al., 2022), and kNN (Ramaswamy et al., 2000) with $k = 5$. We utilize the adjusted best F1 score (Goswami et al., 2023; Challu et al., 2022) and VUS-ROC score (Paparrizos et al., 2022) reported for each baseline by Goswami et al. (2024).

**Forecasting** We further evaluate TiViT in long-horizon time series forecasting on 8 standard datasets (Wu et al., 2021; Ilbert et al., 2024). Similar to the best setup for anomaly detection, TiViT utilizes 6 layers of OpenCLIP ViT-B as backbone, applies no patch overlap, and flattens the sequence of representations. A linear forecasting head is learned per dataset and forecasting horizon in $\{96, 192, 336, 720\}$. Our comparison considers 8 baselines. There are 2 TSFMs evaluated with linear probing: Moment (Goswami et al., 2024) and GPT4TS (Zhou et al., 2023). Moreover, there are 6 supervised methods: PatchTST (Nie et al., 2023), DLinear (Zeng et al., 2023), TimesNet (Wu et al., 2023), FEDformer (Zhou et al., 2022), N-BEATS (Oreshkin et al., 2020), and Stationary. The Mean Squared Error (MSE) and Mean Absolute Error (MAE) per baseline have been reported by Goswami et al. (2024).

*Table 5.* Comparison of interpolation methods on the UCR benchmark.

| Interpolation | Antialias | Accuracy |
|---|---|---|
| Bilinear | False | 81.2 |
| | True | 80.9 |
| Bicubic | False | 79.1 |
| | True | 79.1 |
| Lanczos | - | 80.6 |
| Nearest | - | 81.6 |

## B. Additional Analysis on TiViT

### B.1. Patch Size and Overlap

In Section 4.1, we report for TiViT that a patch size $P = \sqrt{T}$ and a stride $S = \frac{P}{10}$ yields high classification accuracy on any time series of length $T$. The patch size parameter $P$ affects the visual appearance of the image representation provided to the ViT for feature extraction. Figure 5 displays a time series sample from the ECG200 (Olszewski, 2001) dataset along with its corresponding image representations for three different patch sizes. After patching and stacking, the 2D matrix is resized to the quadratic image resolution required by ViTs. Using very small (Figure 5b) or very large (Figure 5d) patch sizes results in redundant tokens representing the same input signal. To avoid a computationally expensive hyperparameter search to find the best patch size $P^*$ per dataset, we propose to select $P = \sqrt{T}$ for any dataset of length $T$. A patch size of $\sqrt{T}$ yields a square-shaped image prior to resizing and thus the most diverse set of patches without any horizontal or vertical distortion (Figure 5c). Moreover, this setting is in line with our theoretical consideration in Section D.1.

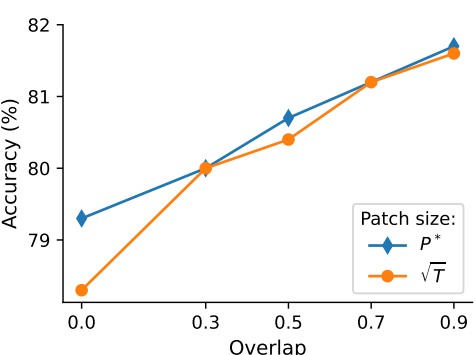

*Figure 6.* Effect of patch size and overlap on classification accuracy.

Table 4 presents the classification accuracy for TiViT with a CLIP backbone (TiViT-CLIP) and both non-overlapping and overlapping patches. To provide an upper bound on the classification performance, we perform a hyperparameter search for the best patch size $P^*$. Specifically, for each dataset of length $T$, we consider 20 equally spaced values in $[1, \frac{T}{2}]$ and identify the patch size that maximizes classification accuracy on the validation set. Note that, while there is a small decline in accuracy in the case of no overlap, when consistently applying $P = \sqrt{T}$, the computational cost is reduced by a factor of 20. The impact of the correct patch size vanishes with increasing overlap.

### B.2. Interpolation Algorithm for Image Resizing

In our time series-to-image transformation, we resize the grayscale images to the resolution expected by the ViT with nearest interpolation by default. To further investigate the impact of the resizing method, we conduct additional experiments using bilinear and bicubic interpolation, both with and without antialiasing, and Lanczos interpolation. Table 5 summarizes our results on the UCR benchmark and indicates that nearest interpolation yields the highest classification accuracy. We hypothesize that nearest interpolation is optimal for TiViT since it preserves the raw time series signals without introducing any smoothing artifacts.

### B.3. Imaging Method for Time Series

In Section 3.1, we describe the transformation of time series into grayscale heatmaps. Here, we explore two alternative image representations. Specifically, we visualize the time series as line plots, similar to (Li et al., 2023b), and Gramian Angular Fields (GAF). We provide these 2D representations to TiViT and evaluate their effectiveness for classification on the UCR benchmark. For the two new imaging methods, we perform a hyperparameter search on the hidden layers ([10, 14, 18]) and choose the best configuration based on validation accuracy. The test accuracy is shown in Table 6. Our results

*Table 6.* Comparison of imaging methods on the UCR benchmark.

| Imaging method | Backbone | Layer | Accuracy |
|---|---|---|---|
| Gramian Angular Field | ViT-H/14 | 14 | 76.4 |
| Lineplot | ViT-H/14 | 14 | 80.7 |
| Heatmap | ViT-H/14 | 14 | 81.6 |

*Table 7.* Linear classification accuracy of TiViT on the UCR dataset with different ways of aggregating the hidden representations per layer. We report the total number of layers including the output layer and the index of the best performing layer starting from 0.

| Model | # Layers | Average of tokens | | CLS token | |
|---|---|---|---|---|---|
| | | Layer | Accuracy | Layer | Accuracy |
| TiViT-DINOv2 | 25 | 15 | 80.0 | 17 | 79.1 |
| TiViT-SigLIP 2 | 28 | 10 | 80.6 | 14 | 71.7 |
| TiViT-CLIP | 33 | 14 | **81.6** | 18 | 78.6 |

indicate that TiViT achieves the highest classification accuracy using the heatmap-based representations.

### B.4. Aggregation of Hidden Representations

As described in Section 3.2, we obtain a single embedding for each time series by averaging the ViT hidden representations in a particular layer. We now evaluate the performance of TiViT when using the CLS token from each layer instead. Table 7 compares the linear classification performance on the UCR dataset using either the CLS token or the mean of all tokens. To ensure a fair comparison, we determine the best performing layer for each approach based on the validation accuracy. Across all backbones, the CLS token consistently results in lower test accuracy, confirming our choice to use the mean hidden representation in TiViT. Interestingly, the best performing CLS tokens appear in later layers compared to the best performing mean tokens. Therefore, utilizing the mean representations not only enhances classification accuracy, but also reduces computational cost.

### B.5. Intrinsic Dimension and Principal Components of Hidden Representations

The intrinsic dimension quantifies the minimum number of variables required to represent a local neighborhood of samples in the representation space. To estimate the intrinsic dimension, the TWO-NN estimator introduced by Facco et al. (2017) leverages the distance of each data point to its first and second nearest neighbor. As noted by the authors, a larger number of data points reduces the average distance to the second neighbor, and thus increases the intrinsic dimension. To mitigate this effect, they propose to subsample the dataset. Given a dataset of size $N$, we report the intrinsic dimension for $\frac{N}{4}$ subsamples in the main paper, which is in line with (Valeriani et al., 2023). In Figure 7, we compare the intrinsic dimension of average representations from hidden layers using $N$, $\frac{N}{2}$, $\frac{N}{4}$, and $\frac{N}{8}$ samples for estimation. The layer with the highest intrinsic

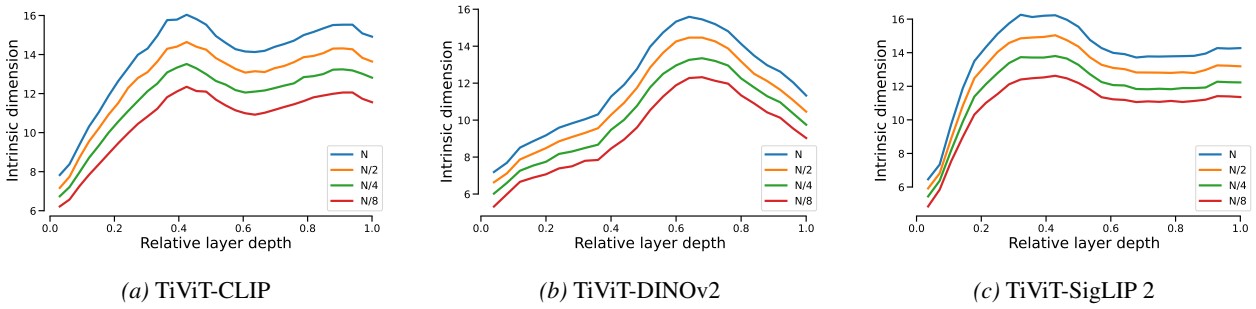

| (a) TiViT-CLIP | (b) TiViT-DINOv2 | (c) TiViT-SigLIP 2 |
|---|---|---|

*Figure 7.* Intrinsic dimension of hidden representations per layer from CLIP, DINOv2, and SigLIP computed for subsamples of the dataset in $\{N, \frac{N}{2}, \frac{N}{4}, \frac{N}{8}\}$.

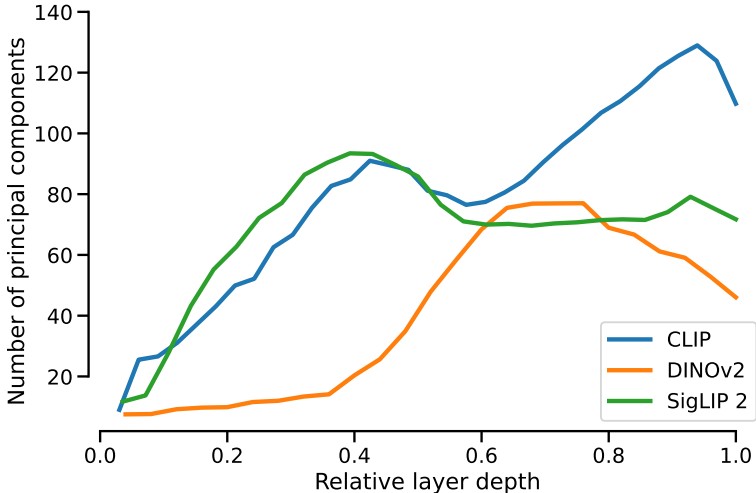

*Figure 8.* Number of principal components necessary to cover 95% of variance in the ViT representations per layer averaged across UCR datasets.

*Table 8.* Linear classification with TiViT on the UCR benchmark. For each model, we report the test accuracy achieved with the best performing hidden layer.

| Model | Architecture | Layer (Max) | Parameters | Data | Accuracy |
|---|---|---|---|---|---|
| TiViT-DINOv3 | ViT-L/14 | 17 (25) | 202 M | LVD-1689M | 80.2 |
| TiViT-SigLIP 2 | SoViT-400m/14 | 12 (28) | 184 M | WebLI (10B) | 80.6 |
| TiViT-CLIP | ViT-H/14 | 14 (33) | 257 M | LAION-2B | **81.6** |

dimension, which is central to our analysis, remains the same regardless of the subsampling ratio.

Since the intrinsic dimension only characterizes the local geometry of the representation space, we further provide a global analysis using principal components. Specifically, in Figure 8, we determine the number of principal components that are necessary to cover 95% of the variance in the data. For DINOv2, we observe a peak in the number of principal components in the middle layers that corresponds to the layers achieving the best classification accuracy. Interestingly, CLIP and SigLIP 2 exhibit two peaks in the number of principal components across the layers. The middle layers corresponding to the first peak yield the highest time series classification accuracy.

### B.6. Size of ViT Backbone

We report the performance of TiViT with CLIP ViT-H backbone in Section 4.2 of the main paper. Table 9 provides a detailed analysis of how the performance of TiViT varies with the size of the ViT backbone, including ViT-B (with two patch sizes), ViT-L, and ViT-H. Remarkably, with only 6 Transformer layers from ViT-B, TiViT achieves an accuracy of 80.8%. While matching the number of Transformer layers in Mantis, TiViT surpasses Mantis (80.1%) in classification accuracy. However, the hidden dimensionality is higher for the ViT-B backbone used in TiViT. By utilizing a larger backbone, specifically 14 hidden layers of ViT-H/14, we achieve the highest accuracy of 81.6%, significantly outperforming conventional TSFMs.

*Table 9.* Linear classification of TiViT-CLIP with varying size of the ViT backbone. For each model, we report the test accuracy on the UCR dataset achieved with the best performing hidden layer representation and the number of parameters up to this layer.

| Architecture | Layer (total number) | Parameters | Accuracy |
|---|---|---|---|
| ViT-B/32 | 8 (13) | 52 M | 79.8 |
| ViT-B/16 | 6 (13) | 36 M | 80.8 |
| ViT-L/14 | 10 (25) | 178 M | 80.3 |
| ViT-H/14 | 14 (33) | 257 M | **81.6** |

*Table 10.* Comparison of CLIP-ViT-L-14 pretraining datasets on UCR benchmark.

| Dataset | Backbone | Layer | Accuracy |
|---|---|---|---|
| Laion-400M | CLIP-ViT-L/14 | 10 | 81.6 |
| Laion-2B | CLIP-ViT-L/14 | 10 | 80.5 |

*Table 11.* Comparison of different backbones and feature extraction layers on the UCR benchmark.

| Backbone | Layer | Accuracy |
|---|---|---|
| ViT-H/14 | 14 | 81.6 |
| ConvNeXt-XXLarge | 15 | 82.1 |

## B.7. Size of Pretraining Dataset

ViTs are pretrained on massive image datasets to learn rich and transferable features. These image datasets are orders of magnitude larger than the time series corpora used to pretrain models such as Mantis (2M samples) or Moment (13M samples). To investigate how the size of the ViT pretraining dataset affects the classification performance of TiViT, we compare TiViT with a CLIP-ViT-L backbone pretrained on 400M and 2B samples. As shown in Table 10, the model pretrained on 400M images outperforms the one pretrained on 2B images in time series classification. This suggests that dataset size alone does not guarantee superior performance in cross-domain tasks.

## B.8. Convolutional Backbone

We focus our study on ViTs because they are the most widely used vision backbones, trained on the largest datasets, and thus enable a comparison of different pretraining paradigms. Nonetheless, we also include a comparison with CNN-based methods. DINOv2, SigLIP 2, and MAE are exclusively built upon ViTs, and thus the only setting we can identify with a convolutional backbone (ConvNeXt) is OpenCLIP. We perform an ablation study for TiViT using different ConvNeXt layers in $\{10, 15, 20, 25\}$ and evaluate the classification accuracy on the UCR benchmark. As shown in Table 11, our method TiViT is fully compatible with pretrained convolutional models and can achieve even higher accuracies on the UCR benchmark when using a ConvNeXt backbone compared to the typical ViT.

## B.9. Masked Autoencoder Backbone

In the main paper, we analyze the reusability of ViT backbones from CLIP (Radford et al., 2021; Schuhmann et al., 2022), DINOv3 (Siméoni et al., 2025), and SigLIP 2 (Tschannen et al., 2025) in time series classification. In contrast, Chen et al. (2024) repurpose Masked Autoencoders (MAEs) (He et al., 2022) for time series forecasting. To enable a direct comparison, we now utilize the hidden representations of MAE Base, Large, and Huge in time series classification.

Our analysis in Table 12 shows that for MAEs using the CLS token yields better performance in time series classification than averaging token representations. Moreover, Table 12 presents a comparison across MAEs of different sizes, showing that larger backbones consistently achieve higher accuracy. Different from contrastively pretrained models, summarized in Table 8, the best representations for time series classification with MAE lie in later layers. We further observe that the hidden representations of the later MAE layers up to the output layer perform similarly in time series classification, while there is a significant gap between hidden representations and output representations for TiViT-CLIP (see Figure 5 in the

*Table 12.* Linear classification accuracy of TiViT with varying MAE backbone size and aggregation of hidden representations per layer. We report the total number of layers including the output layer and the index of the best performing layer starting from 0.

| Architecture | # Layers | Average of tokens | | CLS token | |
|---|---|---|---|---|---|
| | | Layer | Acc | Layer | Acc |
| MAE Base | 13 | 8 | 72.7 | 9 | 73.8 |
| MAE Large | 25 | 14 | 74.3 | 18 | 75.6 |
| MAE Huge | 33 | 20 | 75.9 | 20 | **76.7** |

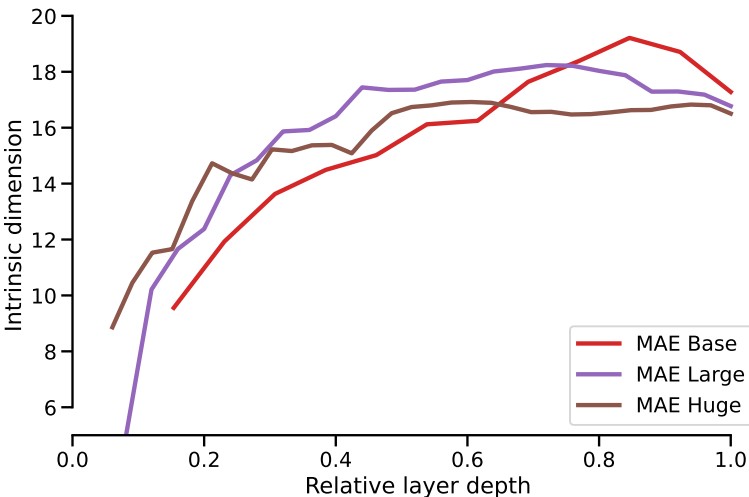

*Figure 9.* Intrinsic dimensionality of CLS tokens per MAE layer averaged across UCR datasets.

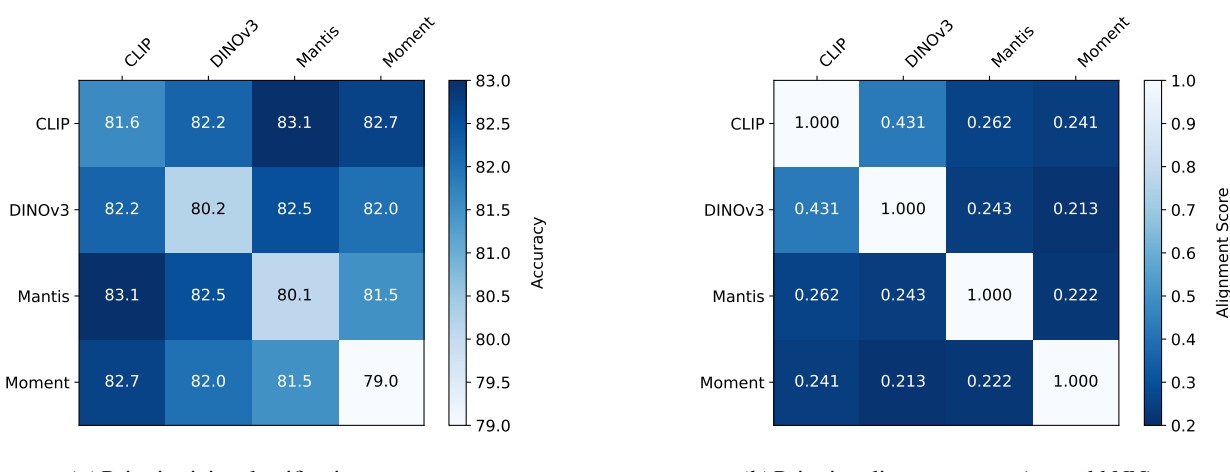

*(a)* Pairwise joint classification accuracy.   *(b)* Pairwise alignment score (mutual kNN).

*Figure 10.* The representations of frozen ViTs and TSFMs are concatenated and used in linear classification. Results are averaged over 128 datasets from the UCR benchmark.

main paper). Figure 9 illustrates the intrinsic dimension of the CLS tokens per layer averaged across the UCR datasets. We observe that the intrinsic dimension increases up to 60% of the layer depth, while the later layers mostly exhibit a similar intrinsic dimension, explaining their similar classification performance.

It is worth noting that MAE has only been pretrained on ImageNet-1k (Deng et al., 2009) with 1.5 million samples, whereas CLIP has been pretrained on the significantly larger LAION-2B (Schuhmann et al., 2022) dataset with 2 billion samples. We hypothesize that being exposed to a larger set of images during training enhances the capacity of a vision model to extract discriminative patterns from 2D time series representations.

### B.10. Representational Alignment of TiViT and TSFM

In Table 2 of our main paper, we report the alignment and joint classification accuracy for TiViT and TSFMs. Figure 10 is an additional visualization of the pairwise scores as heatmaps.

## B.11. Qualitative Feature Analysis

In Section 4.6, we apply attention rollout to two samples from the ECG200 dataset, demonstrating that TiViT attends to salient regions of the time series images. Figure 11 further illustrates this behavior with three examples each from the AllGestureWiimoteX and ElectricDevices datasets, showing the original image, the corresponding attention rollout, and the overlay.

We further employ t-SNE to investigate the structure of the representations extracted by TiViT. Figure 12 presents t-SNE visualizations for 12 additional datasets. The results underscore TiViT's ability to uncover intrinsic cluster structures without access to labels and without being explicitly trained on time series.

Another way of understanding the features learned by ViTs is noise maximization. Ghiasi et al. (2022) have generated images that highly activate a particular feature in ViTs starting from random noise. TiViT applies a frozen backbone and thus utilizes the exact same features of a ViT learned from natural images. Their visualizations underline that ViT-B captures general edges and textures in early layers, and more specialized objects in later layers. Please note that TiViT only uses the first six layers of ViT-B, where there are mostly patterns and less semantic components.

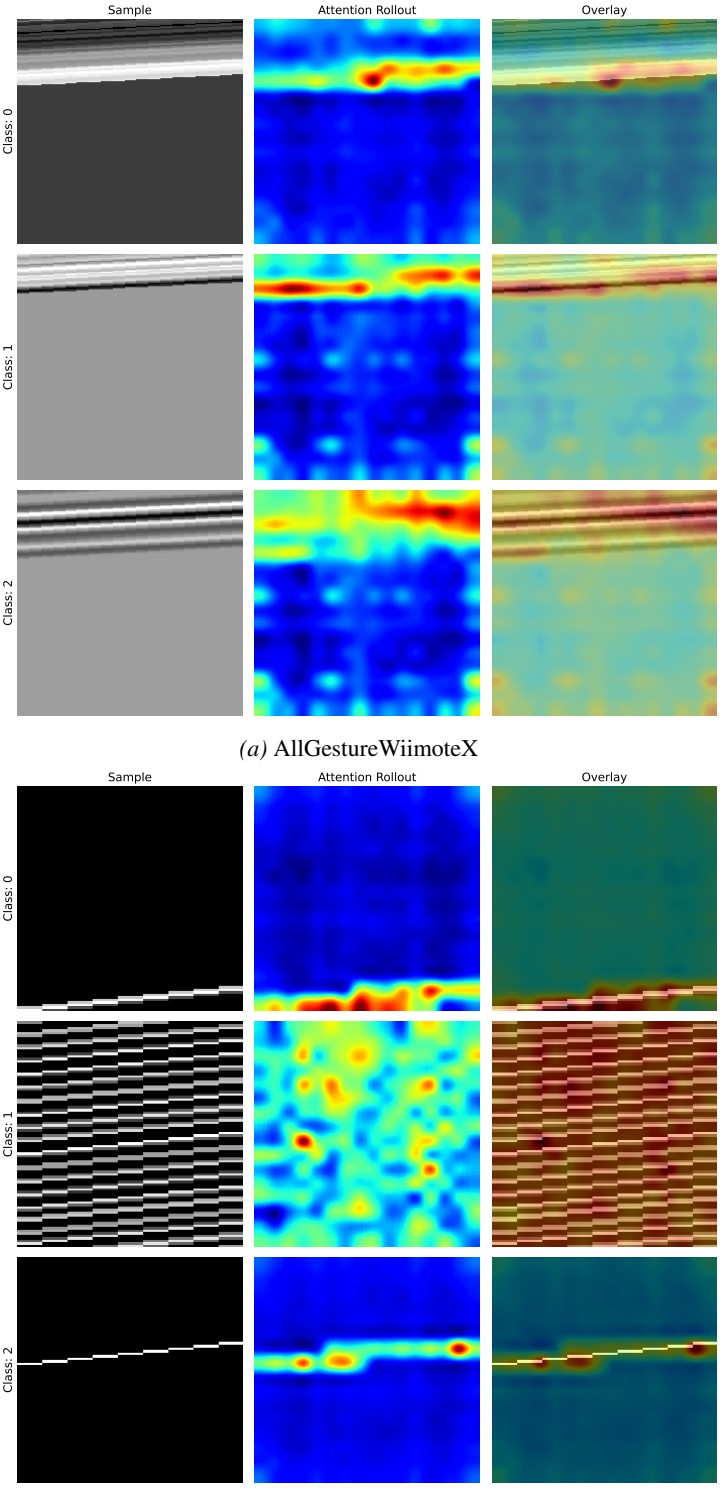

*(a)* AllGestureWiimoteX

*(b)* ElectricDevices

*Figure 11.* Attention rollout.

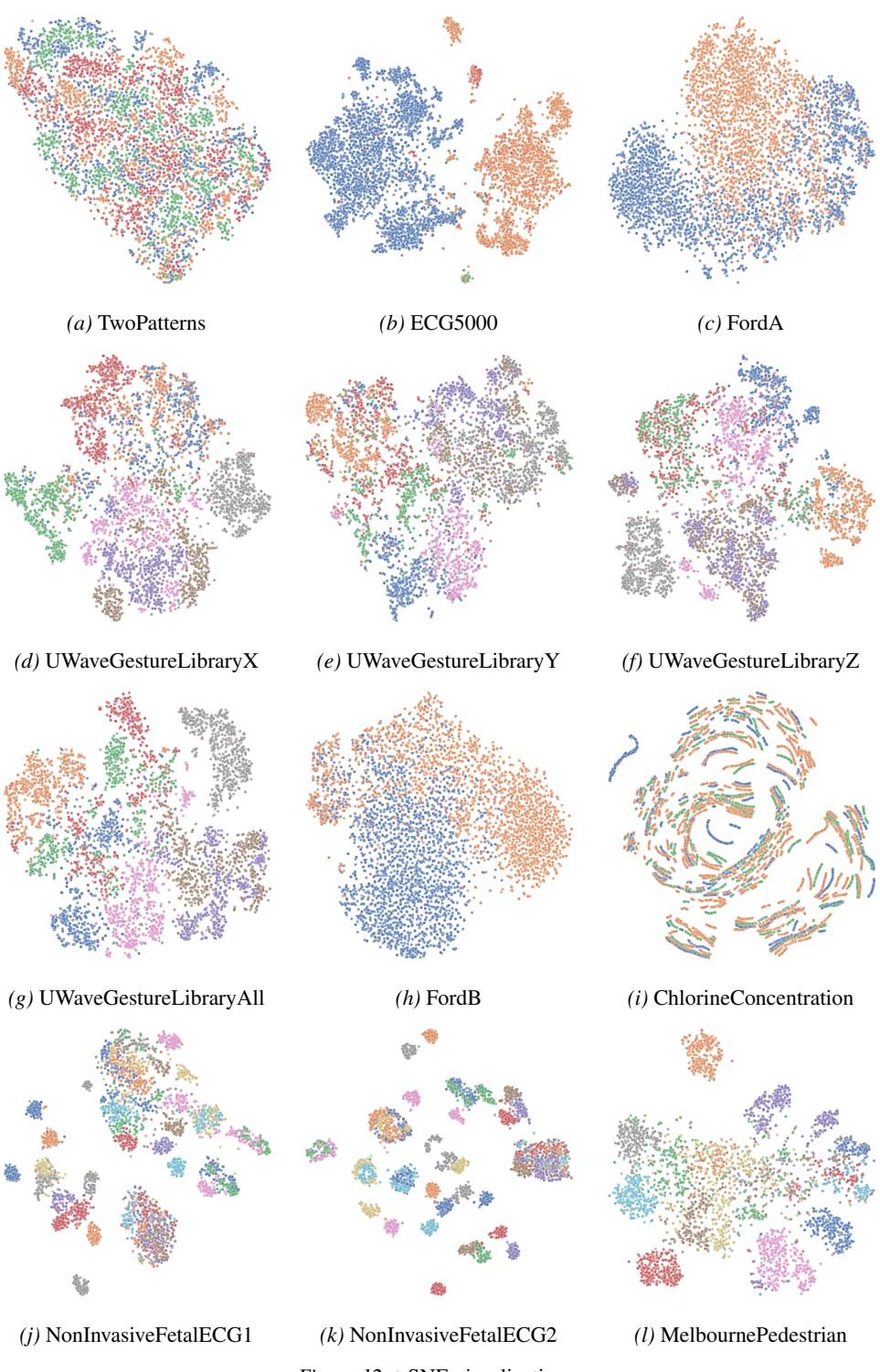

*(a)* TwoPatterns      *(b)* ECG5000      *(c)* FordA

*(d)* UWaveGestureLibraryX      *(e)* UWaveGestureLibraryY      *(f)* UWaveGestureLibraryZ

*(g)* UWaveGestureLibraryAll      *(h)* FordB      *(i)* ChlorineConcentration

*(j)* NonInvasiveFetalECG1      *(k)* NonInvasiveFetalECG2      *(l)* MelbournePedestrian

*Figure 12.* t-SNE visualizations.

## C. Detailed Benchmarking Results

In the main paper, we report the average accuracy of TiViT and TSFM across 128 univariate datasets from the UCR archive and 27 multivariate datasets from the UEA archive. Here, we report the full linear classification benchmark with accuracy scores for Mantis, Moment, TiViT, and their combinations on each dataset. Table 13 presents the performance on the UCR dataset, Table 14 reports the results on the UEA dataset, and Table 15 reports the in-domain and out-of-domain results on the WOODS dataset. Additionally, Table 16 provides the mean rank of all five methods on both benchmarks. If multiple elements share the same rank, we assign them the lowest rank in the group. Comparisons with supervised and self-supervised baselines are provided in Table 17 for the UCR benchmark and in Table 18 for the UEA benchmark.

Furthermore, we assess the performance of TiViT against baseline methods in time series forecasting on 8 standard datasets (Table 19) and in time series anomaly detection on 248 datasets from the UCR Anomaly Archive (Table 20).

*Table 13.* Classification accuracy for 128 univariate datasets from the UCR benchmark. We report the mean and standard deviation across three random seeds.

| Dataset | Moment | Mantis | TiViT | TiViT + Moment | TiViT + Mantis |
|---|---|---|---|---|---|
| ACSF1 | 0.673 ± 0.012 | 0.667 ± 0.021 | **0.773** ± 0.015 | **0.773** ± 0.006 | 0.757 ± 0.015 |
| Adiac | 0.728 ± 0.004 | 0.728 ± 0.011 | 0.708 ± 0.009 | **0.732** ± 0.008 | 0.730 ± 0.012 |
| AllGestureWiimoteX | 0.686 ± 0.010 | 0.699 ± 0.003 | 0.685 ± 0.010 | 0.717 ± 0.009 | **0.726** ± 0.001 |
| AllGestureWiimoteY | 0.710 ± 0.006 | 0.742 ± 0.007 | 0.721 ± 0.015 | 0.750 ± 0.022 | **0.760** ± 0.014 |
| AllGestureWiimoteZ | 0.605 ± 0.007 | 0.673 ± 0.018 | 0.658 ± 0.015 | 0.690 ± 0.014 | **0.700** ± 0.014 |
| ArrowHead | 0.804 ± 0.012 | 0.745 ± 0.007 | 0.819 ± 0.049 | **0.851** ± 0.015 | 0.829 ± 0.035 |
| BME | 0.936 ± 0.010 | 0.991 ± 0.010 | 0.991 ± 0.015 | 0.987 ± 0.018 | **0.996** ± 0.008 |
| Beef | 0.667 ± 0.067 | 0.689 ± 0.019 | **0.800** ± 0.067 | **0.800** ± 0.000 | 0.789 ± 0.069 |
| BeetleFly | 0.850 ± 0.050 | 0.867 ± 0.058 | 0.917 ± 0.058 | 0.917 ± 0.058 | **0.950** ± 0.000 |
| BirdChicken | 0.883 ± 0.029 | **0.950** ± 0.000 | 0.917 ± 0.029 | 0.900 ± 0.000 | 0.933 ± 0.029 |
| CBF | 0.907 ± 0.030 | 0.990 ± 0.009 | **0.999** ± 0.001 | 0.997 ± 0.004 | **0.999** ± 0.001 |
| Car | 0.856 ± 0.035 | 0.828 ± 0.010 | 0.844 ± 0.010 | 0.878 ± 0.010 | **0.889** ± 0.025 |
| Chinatown | 0.962 ± 0.003 | **0.964** ± 0.006 | 0.950 ± 0.018 | 0.954 ± 0.025 | **0.964** ± 0.010 |
| ChlorineConcentration | 0.733 ± 0.010 | 0.643 ± 0.009 | 0.728 ± 0.008 | **0.744** ± 0.012 | 0.738 ± 0.000 |
| CinCECGTorso | 0.719 ± 0.056 | 0.727 ± 0.021 | **0.868** ± 0.034 | 0.837 ± 0.063 | 0.860 ± 0.039 |
| Coffee | **1.000** ± 0.000 | **1.000** ± 0.000 | **1.000** ± 0.000 | **1.000** ± 0.000 | **1.000** ± 0.000 |
| Computers | 0.712 ± 0.036 | 0.740 ± 0.012 | **0.785** ± 0.005 | 0.784 ± 0.011 | 0.781 ± 0.023 |
| CricketX | 0.706 ± 0.020 | 0.726 ± 0.015 | 0.753 ± 0.006 | 0.757 ± 0.013 | **0.765** ± 0.011 |
| CricketY | 0.693 ± 0.018 | 0.732 ± 0.017 | 0.765 ± 0.006 | 0.776 ± 0.008 | **0.783** ± 0.012 |
| CricketZ | 0.740 ± 0.016 | 0.721 ± 0.009 | 0.773 ± 0.017 | 0.779 ± 0.006 | **0.791** ± 0.012 |
| Crop | 0.709 ± 0.003 | 0.695 ± 0.001 | 0.675 ± 0.001 | **0.714** ± 0.003 | 0.707 ± 0.002 |
| DiatomSizeReduction | 0.900 ± 0.030 | 0.881 ± 0.032 | **0.949** ± 0.055 | 0.935 ± 0.048 | 0.944 ± 0.054 |
| DistalPhalanxOutlineAgeGroup | 0.743 ± 0.011 | **0.746** ± 0.017 | 0.703 ± 0.015 | 0.729 ± 0.011 | 0.717 ± 0.011 |
| DistalPhalanxOutlineCorrect | 0.762 ± 0.017 | 0.728 ± 0.007 | **0.769** ± 0.029 | 0.766 ± 0.008 | 0.757 ± 0.014 |
| DistalPhalanxTW | 0.643 ± 0.004 | **0.698** ± 0.017 | 0.640 ± 0.012 | 0.671 ± 0.011 | 0.626 ± 0.019 |
| DodgerLoopDay | 0.442 ± 0.014 | **0.517** ± 0.036 | 0.488 ± 0.043 | 0.467 ± 0.014 | 0.508 ± 0.040 |
| DodgerLoopGame | 0.691 ± 0.062 | 0.720 ± 0.018 | 0.797 ± 0.045 | 0.766 ± 0.073 | **0.802** ± 0.061 |
| DodgerLoopWeekend | **0.986** ± 0.013 | 0.978 ± 0.007 | 0.959 ± 0.011 | 0.981 ± 0.008 | 0.969 ± 0.015 |
| ECG200 | 0.843 ± 0.006 | 0.840 ± 0.017 | **0.863** ± 0.006 | 0.847 ± 0.031 | 0.847 ± 0.021 |
| ECG5000 | 0.934 ± 0.002 | 0.926 ± 0.005 | 0.934 ± 0.002 | **0.936** ± 0.003 | **0.936** ± 0.004 |
| ECGFiveDays | 0.919 ± 0.059 | 0.967 ± 0.012 | 0.953 ± 0.030 | **0.972** ± 0.032 | 0.959 ± 0.028 |
| EOGHorizontalSignal | 0.559 ± 0.012 | 0.542 ± 0.014 | 0.598 ± 0.008 | 0.634 ± 0.008 | **0.642** ± 0.012 |
| EOGVerticalSignal | 0.462 ± 0.021 | **0.530** ± 0.013 | 0.445 ± 0.006 | 0.476 ± 0.016 | 0.471 ± 0.008 |
| Earthquakes | **0.734** ± 0.025 | 0.707 ± 0.018 | 0.698 ± 0.007 | 0.717 ± 0.008 | 0.703 ± 0.017 |
| ElectricDevices | 0.626 ± 0.006 | 0.698 ± 0.003 | **0.757** ± 0.009 | 0.741 ± 0.003 | 0.748 ± 0.007 |
| EthanolLevel | **0.649** ± 0.008 | 0.433 ± 0.004 | 0.574 ± 0.008 | 0.617 ± 0.013 | 0.586 ± 0.008 |
| FaceAll | 0.724 ± 0.006 | **0.797** ± 0.007 | 0.741 ± 0.005 | 0.743 ± 0.005 | 0.762 ± 0.007 |
| FaceFour | 0.826 ± 0.076 | **0.958** ± 0.007 | 0.871 ± 0.029 | 0.909 ± 0.034 | 0.936 ± 0.035 |
| FacesUCR | 0.789 ± 0.010 | 0.888 ± 0.003 | 0.881 ± 0.007 | 0.881 ± 0.004 | **0.912** ± 0.004 |
| FiftyWords | 0.733 ± 0.015 | 0.736 ± 0.010 | 0.758 ± 0.013 | 0.788 ± 0.003 | **0.796** ± 0.006 |
| Fish | 0.949 ± 0.000 | 0.954 ± 0.000 | 0.952 ± 0.007 | 0.945 ± 0.020 | **0.968** ± 0.013 |
| FordA | 0.915 ± 0.002 | 0.910 ± 0.003 | 0.915 ± 0.003 | **0.927** ± 0.004 | 0.917 ± 0.000 |
| FordB | 0.801 ± 0.004 | 0.769 ± 0.002 | **0.812** ± 0.005 | 0.809 ± 0.007 | 0.800 ± 0.012 |
| FreezerRegularTrain | 0.973 ± 0.011 | 0.976 ± 0.012 | **0.997** ± 0.002 | 0.996 ± 0.005 | **0.997** ± 0.002 |
| FreezerSmallTrain | 0.840 ± 0.012 | 0.870 ± 0.020 | **0.992** ± 0.004 | 0.982 ± 0.006 | 0.990 ± 0.003 |
| Fungi | 0.753 ± 0.033 | 0.810 ± 0.025 | 0.787 ± 0.022 | 0.806 ± 0.014 | **0.812** ± 0.023 |
| GestureMidAirD1 | 0.659 ± 0.012 | 0.664 ± 0.027 | 0.746 ± 0.013 | 0.731 ± 0.023 | **0.756** ± 0.032 |
| GestureMidAirD2 | 0.567 ± 0.016 | 0.585 ± 0.040 | 0.667 ± 0.012 | 0.644 ± 0.032 | **0.669** ± 0.015 |
| GestureMidAirD3 | 0.359 ± 0.019 | 0.392 ± 0.013 | **0.472** ± 0.016 | 0.449 ± 0.016 | 0.464 ± 0.025 |
| GesturePebbleZ1 | 0.893 ± 0.015 | 0.917 ± 0.003 | 0.895 ± 0.006 | 0.924 ± 0.000 | **0.928** ± 0.003 |
| GesturePebbleZ2 | 0.846 ± 0.018 | **0.895** ± 0.007 | 0.840 ± 0.010 | 0.861 ± 0.035 | 0.892 ± 0.017 |
| GunPoint | 0.984 ± 0.027 | 0.987 ± 0.007 | **0.996** ± 0.004 | 0.987 ± 0.012 | **0.996** ± 0.004 |
| GunPointAgeSpan | 0.980 ± 0.008 | **0.998** ± 0.002 | 0.992 ± 0.002 | 0.993 ± 0.002 | 0.994 ± 0.000 |
| GunPointMaleVersusFemale | **1.000** ± 0.000 | 0.999 ± 0.004 | 0.996 ± 0.002 | **1.000** ± 0.000 | **1.000** ± 0.000 |
| GunPointOldVersusYoung | **1.000** ± 0.000 | **1.000** ± 0.000 | 0.988 ± 0.002 | **1.000** ± 0.000 | **1.000** ± 0.000 |
| Ham | **0.752** ± 0.025 | 0.667 ± 0.010 | 0.695 ± 0.000 | 0.721 ± 0.024 | 0.724 ± 0.019 |
| HandOutlines | 0.930 ± 0.007 | 0.931 ± 0.006 | 0.936 ± 0.007 | **0.945** ± 0.010 | 0.932 ± 0.007 |
| Haptics | 0.491 ± 0.026 | 0.462 ± 0.002 | 0.498 ± 0.007 | 0.535 ± 0.040 | **0.539** ± 0.009 |
| Herring | **0.698** ± 0.018 | 0.682 ± 0.024 | 0.599 ± 0.009 | 0.630 ± 0.039 | 0.625 ± 0.027 |
| HouseTwenty | 0.947 ± 0.010 | 0.961 ± 0.010 | 0.972 ± 0.005 | 0.972 ± 0.010 | **0.980** ± 0.005 |
| InlineSkate | 0.364 ± 0.019 | 0.334 ± 0.021 | 0.398 ± 0.015 | 0.401 ± 0.006 | **0.408** ± 0.015 |
| InsectEPGRegularTrain | 0.987 ± 0.014 | **1.000** ± 0.000 | **1.000** ± 0.000 | **1.000** ± 0.000 | **1.000** ± 0.000 |
| InsectEPGSmallTrain | 0.953 ± 0.008 | **1.000** ± 0.000 | 0.968 ± 0.007 | 0.973 ± 0.005 | 0.999 ± 0.002 |
| InsectWingbeatSound | 0.539 ± 0.003 | 0.470 ± 0.019 | 0.536 ± 0.015 | **0.560** ± 0.007 | 0.539 ± 0.010 |
| ItalyPowerDemand | **0.938** ± 0.005 | 0.910 ± 0.006 | 0.920 ± 0.018 | 0.936 ± 0.011 | 0.923 ± 0.018 |
| LargeKitchenAppliances | 0.859 ± 0.005 | 0.820 ± 0.010 | **0.883** ± 0.014 | 0.873 ± 0.018 | 0.879 ± 0.014 |

Continuation of Table 13

| Dataset | Moment | Mantis | TiViT | TiViT + Moment | TiViT + Mantis |
|---|---|---|---|---|---|
| Lightning2 | 0.760 ± 0.041 | 0.781 ± 0.025 | 0.803 ± 0.028 | **0.820** ± 0.028 | 0.803 ± 0.016 |
| Lightning7 | 0.836 ± 0.036 | 0.749 ± 0.021 | 0.831 ± 0.021 | **0.881** ± 0.008 | 0.822 ± 0.024 |
| Mallat | 0.915 ± 0.010 | 0.868 ± 0.028 | 0.956 ± 0.017 | **0.963** ± 0.016 | 0.958 ± 0.018 |
| Meat | 0.911 ± 0.038 | **0.939** ± 0.019 | 0.800 ± 0.000 | 0.900 ± 0.029 | 0.850 ± 0.044 |
| MedicalImages | 0.730 ± 0.003 | 0.707 ± 0.024 | 0.740 ± 0.006 | **0.780** ± 0.006 | 0.761 ± 0.014 |
| MelbournePedestrian | **0.933** ± 0.003 | 0.908 ± 0.005 | 0.862 ± 0.006 | 0.932 ± 0.005 | 0.925 ± 0.003 |
| MiddlePhalanxOutlineAgeGroup | 0.489 ± 0.029 | **0.587** ± 0.019 | 0.537 ± 0.036 | 0.530 ± 0.004 | 0.571 ± 0.023 |
| MiddlePhalanxOutlineCorrect | 0.816 ± 0.009 | **0.845** ± 0.009 | 0.789 ± 0.015 | 0.792 ± 0.016 | 0.805 ± 0.016 |
| MiddlePhalanxTW | 0.506 ± 0.019 | 0.442 ± 0.017 | 0.506 ± 0.023 | 0.498 ± 0.025 | **0.511** ± 0.010 |
| MixedShapesRegularTrain | 0.947 ± 0.004 | 0.955 ± 0.006 | 0.974 ± 0.002 | 0.973 ± 0.003 | **0.976** ± 0.002 |
| MixedShapesSmallTrain | 0.882 ± 0.004 | 0.904 ± 0.002 | 0.950 ± 0.002 | 0.937 ± 0.004 | **0.957** ± 0.003 |
| MoteStrain | 0.889 ± 0.028 | 0.895 ± 0.026 | 0.875 ± 0.021 | **0.918** ± 0.008 | 0.901 ± 0.025 |
| NonInvasiveFetalECGThorax1 | 0.919 ± 0.002 | 0.797 ± 0.002 | 0.884 ± 0.004 | **0.924** ± 0.003 | 0.885 ± 0.009 |
| NonInvasiveFetalECGThorax2 | 0.927 ± 0.002 | 0.817 ± 0.004 | 0.915 ± 0.001 | **0.934** ± 0.004 | 0.918 ± 0.005 |
| OSULeaf | 0.917 ± 0.004 | 0.899 ± 0.005 | 0.977 ± 0.006 | 0.972 ± 0.010 | **0.978** ± 0.009 |
| OliveOil | **0.856** ± 0.051 | 0.822 ± 0.107 | 0.656 ± 0.077 | 0.778 ± 0.019 | 0.711 ± 0.051 |
| PLAID | 0.775 ± 0.017 | 0.852 ± 0.001 | 0.888 ± 0.008 | 0.901 ± 0.011 | **0.928** ± 0.012 |
| PhalangesOutlinesCorrect | **0.795** ± 0.006 | 0.794 ± 0.008 | 0.789 ± 0.004 | **0.795** ± 0.008 | 0.787 ± 0.004 |
| Phoneme | 0.277 ± 0.003 | 0.293 ± 0.004 | 0.377 ± 0.006 | 0.372 ± 0.003 | **0.386** ± 0.006 |
| PickupGestureWiimoteZ | 0.713 ± 0.042 | 0.767 ± 0.023 | 0.887 ± 0.031 | 0.847 ± 0.046 | **0.893** ± 0.023 |
| PigAirwayPressure | 0.109 ± 0.007 | 0.588 ± 0.012 | 0.540 ± 0.006 | 0.447 ± 0.013 | **0.598** ± 0.010 |
| PigArtPressure | 0.780 ± 0.011 | 0.827 ± 0.017 | 0.817 ± 0.013 | 0.833 ± 0.019 | **0.846** ± 0.005 |
| PigCVP | 0.747 ± 0.027 | 0.753 ± 0.007 | 0.702 ± 0.019 | 0.761 ± 0.018 | **0.801** ± 0.012 |
| Plane | 0.997 ± 0.005 | **1.000** ± 0.000 | **1.000** ± 0.000 | **1.000** ± 0.000 | **1.000** ± 0.000 |
| PowerCons | 0.931 ± 0.006 | 0.933 ± 0.010 | 0.894 ± 0.022 | **0.943** ± 0.013 | 0.906 ± 0.020 |
| ProximalPhalanxOutlineAgeGroup | 0.802 ± 0.020 | **0.852** ± 0.007 | 0.833 ± 0.027 | 0.824 ± 0.005 | 0.828 ± 0.017 |
| ProximalPhalanxOutlineCorrect | 0.883 ± 0.010 | **0.885** ± 0.008 | 0.861 ± 0.020 | 0.871 ± 0.016 | 0.858 ± 0.023 |
| ProximalPhalanxTW | **0.767** ± 0.010 | 0.740 ± 0.015 | 0.751 ± 0.022 | 0.730 ± 0.010 | 0.759 ± 0.023 |
| RefrigerationDevices | 0.496 ± 0.017 | 0.526 ± 0.022 | 0.555 ± 0.007 | 0.531 ± 0.005 | **0.570** ± 0.014 |
| Rock | 0.727 ± 0.031 | 0.700 ± 0.060 | **0.873** ± 0.099 | **0.873** ± 0.115 | 0.853 ± 0.117 |
| ScreenType | 0.499 ± 0.020 | 0.468 ± 0.026 | 0.530 ± 0.014 | 0.516 ± 0.002 | **0.552** ± 0.027 |
| SemgHandGenderCh2 | 0.761 ± 0.018 | 0.883 ± 0.006 | 0.879 ± 0.001 | 0.878 ± 0.013 | **0.914** ± 0.006 |
| SemgHandMovementCh2 | 0.398 ± 0.010 | 0.654 ± 0.018 | 0.545 ± 0.016 | 0.538 ± 0.031 | **0.688** ± 0.024 |
| SemgHandSubjectCh2 | 0.648 ± 0.013 | 0.826 ± 0.005 | 0.840 ± 0.002 | 0.838 ± 0.012 | **0.895** ± 0.007 |
| ShakeGestureWiimoteZ | 0.887 ± 0.012 | 0.867 ± 0.012 | 0.827 ± 0.031 | **0.907** ± 0.031 | 0.840 ± 0.020 |
| ShapeletSim | 0.967 ± 0.010 | 0.919 ± 0.012 | **1.000** ± 0.000 | **1.000** ± 0.000 | **1.000** ± 0.000 |
| ShapesAll | 0.886 ± 0.003 | 0.844 ± 0.010 | 0.901 ± 0.003 | **0.913** ± 0.008 | 0.908 ± 0.007 |
| SmallKitchenAppliances | 0.733 ± 0.010 | 0.796 ± 0.013 | **0.830** ± 0.003 | 0.817 ± 0.018 | 0.812 ± 0.008 |
| SmoothSubspace | 0.898 ± 0.023 | **0.971** ± 0.004 | 0.956 ± 0.010 | 0.964 ± 0.010 | **0.971** ± 0.010 |
| SonyAIBORobotSurface1 | 0.834 ± 0.013 | 0.858 ± 0.015 | 0.890 ± 0.012 | 0.869 ± 0.009 | **0.896** ± 0.010 |
| SonyAIBORobotSurface2 | 0.855 ± 0.027 | 0.895 ± 0.012 | 0.911 ± 0.049 | 0.914 ± 0.049 | **0.923** ± 0.048 |
| StarLightCurves | 0.969 ± 0.003 | 0.968 ± 0.002 | 0.973 ± 0.002 | **0.976** ± 0.002 | **0.976** ± 0.002 |
| Strawberry | **0.972** ± 0.002 | 0.960 ± 0.004 | 0.959 ± 0.002 | 0.968 ± 0.006 | 0.959 ± 0.003 |
| SwedishLeaf | 0.915 ± 0.007 | 0.942 ± 0.006 | 0.955 ± 0.003 | **0.959** ± 0.006 | 0.958 ± 0.003 |
| Symbols | 0.957 ± 0.019 | 0.957 ± 0.031 | 0.966 ± 0.034 | **0.973** ± 0.020 | 0.967 ± 0.035 |
| SyntheticControl | 0.966 ± 0.004 | 0.992 ± 0.002 | 0.999 ± 0.002 | 0.993 ± 0.003 | **1.000** ± 0.000 |
| ToeSegmentation1 | **0.963** ± 0.007 | 0.952 ± 0.012 | 0.952 ± 0.012 | **0.963** ± 0.005 | 0.959 ± 0.009 |
| ToeSegmentation2 | 0.885 ± 0.015 | **0.954** ± 0.008 | 0.923 ± 0.008 | 0.895 ± 0.027 | 0.926 ± 0.004 |
| Trace | **1.000** ± 0.000 | **1.000** ± 0.000 | **1.000** ± 0.000 | **1.000** ± 0.000 | **1.000** ± 0.000 |
| TwoLeadECG | 0.901 ± 0.020 | 0.998 ± 0.002 | 0.997 ± 0.001 | 0.997 ± 0.001 | **1.000** ± 0.000 |
| TwoPatterns | 0.989 ± 0.001 | 0.946 ± 0.007 | 0.998 ± 0.000 | **0.999** ± 0.001 | 0.998 ± 0.001 |
| UMD | **0.993** ± 0.000 | **0.993** ± 0.000 | **0.993** ± 0.000 | **0.993** ± 0.000 | **0.993** ± 0.000 |
| UWaveGestureLibraryAll | 0.923 ± 0.002 | 0.874 ± 0.004 | 0.940 ± 0.001 | **0.950** ± 0.005 | 0.944 ± 0.003 |
| UWaveGestureLibraryX | 0.792 ± 0.001 | 0.779 ± 0.004 | 0.828 ± 0.004 | **0.838** ± 0.004 | **0.838** ± 0.002 |
| UWaveGestureLibraryY | 0.711 ± 0.006 | 0.678 ± 0.009 | 0.749 ± 0.004 | 0.758 ± 0.004 | **0.763** ± 0.006 |
| UWaveGestureLibraryZ | 0.731 ± 0.001 | 0.742 ± 0.009 | 0.770 ± 0.003 | 0.772 ± 0.004 | **0.786** ± 0.001 |
| Wafer | 0.992 ± 0.002 | 0.996 ± 0.000 | **1.000** ± 0.000 | **1.000** ± 0.000 | **1.000** ± 0.000 |
| Wine | **0.889** ± 0.019 | 0.796 ± 0.037 | 0.599 ± 0.065 | 0.747 ± 0.028 | 0.759 ± 0.049 |
| WordSynonyms | 0.655 ± 0.003 | 0.626 ± 0.017 | 0.649 ± 0.007 | **0.690** ± 0.005 | 0.681 ± 0.006 |
| Worms | 0.745 ± 0.033 | 0.710 ± 0.033 | 0.762 ± 0.027 | **0.805** ± 0.026 | 0.762 ± 0.052 |
| WormsTwoClass | 0.775 ± 0.037 | 0.745 ± 0.040 | 0.784 ± 0.020 | **0.792** ± 0.026 | 0.766 ± 0.022 |
| Yoga | 0.833 ± 0.008 | 0.771 ± 0.014 | 0.826 ± 0.009 | **0.852** ± 0.007 | 0.844 ± 0.007 |

End of Table

*Table 14.* Classification accuracy for 27 multivariate datasets from the UEA benchmark. We report the mean and standard deviation across three random seeds.

| Dataset | Moment | Mantis | TiViT | TiViT + Moment | TiViT + Mantis |
|---|---|---|---|---|---|
| ArticularyWordRecognition | $0.988 \pm 0.002$ | $\mathbf{0.991} \pm 0.002$ | $0.977 \pm 0.003$ | $0.977 \pm 0.003$ | $0.974 \pm 0.005$ |
| BasicMotions | $\mathbf{1.000} \pm 0.000$ | $\mathbf{1.000} \pm 0.000$ | $\mathbf{1.000} \pm 0.000$ | $\mathbf{1.000} \pm 0.000$ | $\mathbf{1.000} \pm 0.000$ |
| CharacterTrajectories | $\mathbf{0.982} \pm 0.001$ | $0.973 \pm 0.001$ | $0.964 \pm 0.005$ | $\mathbf{0.982} \pm 0.001$ | $0.978 \pm 0.005$ |
| Cricket | $\mathbf{1.000} \pm 0.000$ | $0.986 \pm 0.000$ | $\mathbf{1.000} \pm 0.000$ | $\mathbf{1.000} \pm 0.000$ | $\mathbf{1.000} \pm 0.000$ |
| DuckDuckGeese | $\mathbf{0.467} \pm 0.081$ | $0.433 \pm 0.023$ | $0.393 \pm 0.081$ | $0.413 \pm 0.064$ | $0.433 \pm 0.050$ |
| ERing | $0.895 \pm 0.022$ | $0.905 \pm 0.025$ | $0.975 \pm 0.014$ | $0.977 \pm 0.006$ | $\mathbf{0.981} \pm 0.007$ |
| EigenWorms | $0.746 \pm 0.022$ | $0.746 \pm 0.016$ | $\mathbf{0.911} \pm 0.016$ | $0.880 \pm 0.009$ | $\mathbf{0.911} \pm 0.012$ |
| Epilepsy | $\mathbf{1.000} \pm 0.000$ | $0.990 \pm 0.004$ | $\mathbf{1.000} \pm 0.000$ | $\mathbf{1.000} \pm 0.000$ | $\mathbf{1.000} \pm 0.000$ |
| EthanolConcentration | $0.445 \pm 0.013$ | $0.269 \pm 0.044$ | $\mathbf{0.485} \pm 0.012$ | $0.473 \pm 0.030$ | $0.465 \pm 0.019$ |
| FaceDetection | $0.584 \pm 0.007$ | $0.592 \pm 0.006$ | $0.598 \pm 0.004$ | $0.584 \pm 0.007$ | $\mathbf{0.607} \pm 0.005$ |
| FingerMovements | $\mathbf{0.633} \pm 0.045$ | $0.593 \pm 0.025$ | $0.517 \pm 0.040$ | $0.620 \pm 0.036$ | $0.553 \pm 0.050$ |
| HandMovementDirection | $\mathbf{0.279} \pm 0.051$ | $0.212 \pm 0.021$ | $0.275 \pm 0.016$ | $0.257 \pm 0.036$ | $0.257 \pm 0.027$ |
| Handwriting | $0.296 \pm 0.018$ | $\mathbf{0.425} \pm 0.013$ | $0.307 \pm 0.034$ | $0.340 \pm 0.002$ | $0.385 \pm 0.021$ |
| Heartbeat | $0.735 \pm 0.007$ | $\mathbf{0.800} \pm 0.017$ | $0.732 \pm 0.008$ | $0.717 \pm 0.022$ | $0.769 \pm 0.003$ |
| InsectWingbeat | $0.231 \pm 0.012$ | $\mathbf{0.573} \pm 0.017$ | $0.355 \pm 0.008$ | $0.332 \pm 0.018$ | $0.443 \pm 0.020$ |
| JapaneseVowels | $0.918 \pm 0.006$ | $\mathbf{0.978} \pm 0.003$ | $0.940 \pm 0.002$ | $0.938 \pm 0.012$ | $0.933 \pm 0.008$ |
| LSST | $0.571 \pm 0.005$ | $0.607 \pm 0.009$ | $0.604 \pm 0.005$ | $0.610 \pm 0.009$ | $\mathbf{0.652} \pm 0.003$ |
| Libras | $0.861 \pm 0.017$ | $0.887 \pm 0.026$ | $0.907 \pm 0.016$ | $\mathbf{0.922} \pm 0.022$ | $0.920 \pm 0.018$ |
| MotorImagery | $0.530 \pm 0.026$ | $\mathbf{0.563} \pm 0.012$ | $\mathbf{0.563} \pm 0.049$ | $0.560 \pm 0.044$ | $0.553 \pm 0.042$ |
| NATOPS | $0.900 \pm 0.029$ | $\mathbf{0.931} \pm 0.014$ | $0.869 \pm 0.006$ | $0.889 \pm 0.006$ | $0.878 \pm 0.006$ |
| PEMS-SF | $0.705 \pm 0.029$ | $\mathbf{0.788} \pm 0.029$ | $0.709 \pm 0.084$ | $0.763 \pm 0.044$ | $0.742 \pm 0.087$ |
| PhonemeSpectra | $0.186 \pm 0.004$ | $0.272 \pm 0.006$ | $0.245 \pm 0.007$ | $0.265 \pm 0.007$ | $\mathbf{0.286} \pm 0.008$ |
| RacketSports | $0.829 \pm 0.007$ | $\mathbf{0.919} \pm 0.004$ | $0.846 \pm 0.010$ | $0.871 \pm 0.008$ | $0.879 \pm 0.027$ |
| SelfRegulationSCP1 | $0.762 \pm 0.010$ | $0.825 \pm 0.022$ | $0.858 \pm 0.008$ | $0.840 \pm 0.003$ | $\mathbf{0.891} \pm 0.010$ |
| SelfRegulationSCP2 | $0.509 \pm 0.031$ | $0.491 \pm 0.018$ | $\mathbf{0.526} \pm 0.038$ | $0.506 \pm 0.017$ | $0.517 \pm 0.020$ |
| SpokenArabicDigits | $\mathbf{0.981} \pm 0.003$ | $0.907 \pm 0.006$ | $0.969 \pm 0.001$ | $0.979 \pm 0.003$ | $0.972 \pm 0.002$ |
| UWaveGestureLibrary | $0.846 \pm 0.010$ | $0.879 \pm 0.015$ | $0.910 \pm 0.005$ | $0.902 \pm 0.004$ | $\mathbf{0.919} \pm 0.009$ |

*Table 15.* Classification accuracy on EEG datasets from WOODS.

*(a)* In-domain

| Model | PCL | CAP | SEDFx | Mean |
|---|---|---|---|---|
| Mantis | 55.1 | 77.2 | 78.9 | 70.4 |
| Moment | 55.6 | 76.4 | 78.2 | 70.1 |
| TiViT | 55.3 | 81.1 | 80.7 | 72.4 |
| TiViT + Mantis | 57.3 | 82.2 | 81.5 | 73.7 |
| TiViT + Moment | 58.2 | 82.1 | 80.1 | 73.5 |

*(b)* Out-of-domain

| Model | PCL | CAP | SEDFx | Mean |
|---|---|---|---|---|
| Mantis | 53.4 | 65.2 | 76.5 | 65.0 |
| Moment | 54.1 | 70.5 | 75.6 | 66.7 |
| TiViT | 54.7 | 70.5 | 77.8 | 67.7 |
| TiViT + Mantis | 55.7 | 70.0 | 78.0 | 67.9 |
| TiViT + Moment | 55.5 | 72.8 | 76.7 | 68.3 |

*Table 16.* Mean rank of TiViT and TSFMs across datasets from the UCR and UEA archive.

| Model | UCR | UEA |
|---|---|---|
| Moment | 3.75 | 3.33 |
| Mantis | 3.43 | 2.85 |
| TiViT *(Ours)* | 2.97 | 2.85 |
| TiViT + Moment *(Ours)* | 2.20 | 2.63 |
| TiViT + Mantis *(Ours)* | **1.95** | **2.22** |

*Table 17.* Classification accuracy across 91 UCR datasets. Baselines from (Goswami et al., 2024).

| Accuracy | TiViT + Mantis | TiViT | Mantis | MOMENT | TimesNet | GPT4TS | TS2Vec | T-Loss | TNC | TS-TCC |
|---|---|---|---|---|---|---|---|---|---|---|
| Mean | 0.848 | 0.834 | 0.826 | 0.794 | 0.572 | 0.566 | **0.851** | 0.833 | 0.786 | 0.793 |
| Median | **0.880** | 0.849 | 0.852 | 0.815 | 0.565 | 0.583 | 0.871 | 0.849 | 0.788 | 0.802 |
| Std. | 0.133 | 0.136 | 0.143 | 0.147 | 0.238 | 0.234 | 0.134 | 0.136 | 0.168 | 0.176 |

| Accuracy | TST | CNN | Encoder | FCN | MCNN | MLP | ResNet | t-LeNet | TWIESN | DTW |
|---|---|---|---|---|---|---|---|---|---|---|
| Mean | 0.658 | 0.751 | 0.743 | 0.809 | 0.702 | 0.750 | 0.825 | 0.348 | 0.726 | 0.764 |
| Median | 0.720 | 0.773 | 0.753 | 0.837 | 0.718 | 0.766 | 0.852 | 0.333 | 0.724 | 0.768 |
| Std. | 0.220 | 0.180 | 0.159 | 0.188 | 0.194 | 0.169 | 0.177 | 0.221 | 0.164 | 0.152 |

*Table 18.* Classification accuracy across 29 UEA datasets. Baselines from (Goswami et al., 2024).

| Accuracy | TiViT + Mantis | TiViT | Mantis | MOMENT | TS2Vec | T-Loss | TNC | TS-TCC | TST | DTW |
|---|---|---|---|---|---|---|---|---|---|---|
| Mean | **0.719** | 0.706 | 0.693 | 0.670 | 0.694 | 0.646 | 0.660 | 0.657 | 0.605 | 0.638 |
| Median | **0.823** | 0.789 | 0.788 | 0.722 | 0.683 | 0.676 | 0.746 | 0.751 | 0.620 | 0.664 |
| Std. | 0.260 | 0.266 | 0.266 | 0.274 | 0.255 | 0.296 | 0.267 | 0.263 | 0.294 | 0.296 |

*Table 19.* Long-term forecasting. Baselines from (Goswami et al., 2024).

| Method | | Pretraining + linear probing | | | | | | Supervised training | | | | | | | | | | |
|---|---|---|---|---|---|---|---|---|---|---|---|---|---|---|---|---|---|---|
| | | TiViT (Ours) | | MOMENT | | GPT4TS | | PatchTST | | DLinear | | TimesNet | | FEDFormer | | Stationary | | N-BEATS | |
| Metric | | MSE | MAE | MSE | MAE | MSE | MAE | MSE | MAE | MSE | MAE | MSE | MAE | MSE | MAE | MSE | MAE | MSE | MAE |
| Weather | 96 | 0.153 | 0.211 | 0.154 | 0.209 | 0.162 | 0.212 | 0.149 | 0.198 | 0.176 | 0.237 | 0.172 | 0.220 | 0.217 | 0.296 | 0.173 | 0.223 | 0.152 | 0.210 |
| | 192 | 0.196 | 0.247 | 0.197 | 0.248 | 0.204 | 0.248 | 0.194 | 0.241 | 0.220 | 0.282 | 0.219 | 0.261 | 0.276 | 0.336 | 0.245 | 0.285 | 0.199 | 0.260 |
| | 336 | 0.248 | 0.285 | 0.246 | 0.285 | 0.254 | 0.286 | 0.245 | 0.282 | 0.265 | 0.319 | 0.280 | 0.306 | 0.339 | 0.380 | 0.321 | 0.338 | 0.258 | 0.311 |
| | 720 | 0.321 | 0.337 | 0.315 | 0.336 | 0.326 | 0.337 | 0.314 | 0.334 | 0.333 | 0.362 | 0.365 | 0.359 | 0.403 | 0.428 | 0.414 | 0.410 | 0.331 | 0.359 |
| ECL | 96 | 0.140 | 0.240 | 0.136 | 0.233 | 0.139 | 0.238 | 0.129 | 0.222 | 0.140 | 0.237 | 0.168 | 0.272 | 0.193 | 0.308 | 0.169 | 0.273 | 0.131 | 0.228 |
| | 192 | 0.152 | 0.251 | 0.152 | 0.247 | 0.153 | 0.251 | 0.157 | 0.240 | 0.153 | 0.249 | 0.184 | 0.289 | 0.201 | 0.315 | 0.182 | 0.286 | 0.153 | 0.248 |
| | 336 | 0.168 | 0.267 | 0.167 | 0.264 | 0.169 | 0.266 | 0.163 | 0.259 | 0.169 | 0.267 | 0.198 | 0.300 | 0.214 | 0.329 | 0.200 | 0.304 | 0.170 | 0.267 |
| | 720 | 0.204 | 0.297 | 0.205 | 0.295 | 0.206 | 0.297 | 0.197 | 0.290 | 0.203 | 0.301 | 0.220 | 0.320 | 0.246 | 0.355 | 0.222 | 0.321 | 0.208 | 0.298 |
| Traffic | 96 | 0.384 | 0.274 | 0.391 | 0.282 | 0.388 | 0.282 | 0.360 | 0.249 | 0.410 | 0.282 | 0.593 | 0.321 | 0.587 | 0.366 | 0.612 | 0.338 | 0.375 | 0.259 |
| | 192 | 0.398 | 0.280 | 0.404 | 0.287 | 0.407 | 0.290 | 0.379 | 0.256 | 0.423 | 0.287 | 0.617 | 0.336 | 0.604 | 0.373 | 0.613 | 0.340 | 0.403 | 0.274 |
| | 336 | 0.407 | 0.285 | 0.414 | 0.292 | 0.412 | 0.294 | 0.392 | 0.264 | 0.436 | 0.296 | 0.629 | 0.336 | 0.621 | 0.383 | 0.618 | 0.328 | 0.426 | 0.285 |
| | 720 | 0.443 | 0.303 | 0.450 | 0.310 | 0.450 | 0.312 | 0.432 | 0.286 | 0.466 | 0.315 | 0.640 | 0.350 | 0.626 | 0.382 | 0.653 | 0.355 | 0.508 | 0.335 |
| ETTh1 | 96 | 0.391 | 0.417 | 0.387 | 0.410 | 0.376 | 0.397 | 0.370 | 0.399 | 0.375 | 0.399 | 0.384 | 0.402 | 0.376 | 0.419 | 0.513 | 0.491 | 0.399 | 0.428 |
| | 192 | 0.411 | 0.430 | 0.410 | 0.426 | 0.416 | 0.418 | 0.413 | 0.421 | 0.405 | 0.416 | 0.436 | 0.429 | 0.420 | 0.448 | 0.534 | 0.504 | 0.451 | 0.464 |
| | 336 | 0.425 | 0.442 | 0.422 | 0.437 | 0.442 | 0.433 | 0.422 | 0.436 | 0.439 | 0.443 | 0.491 | 0.469 | 0.459 | 0.465 | 0.588 | 0.535 | 0.498 | 0.500 |
| | 720 | 0.447 | 0.469 | 0.454 | 0.472 | 0.477 | 0.456 | 0.447 | 0.466 | 0.472 | 0.490 | 0.521 | 0.500 | 0.506 | 0.507 | 0.643 | 0.616 | 0.608 | 0.573 |
| ETTh2 | 96 | 0.319 | 0.375 | 0.288 | 0.345 | 0.285 | 0.342 | 0.274 | 0.336 | 0.289 | 0.353 | 0.340 | 0.374 | 0.358 | 0.397 | 0.476 | 0.458 | 0.327 | 0.387 |
| | 192 | 0.363 | 0.406 | 0.349 | 0.386 | 0.354 | 0.389 | 0.339 | 0.379 | 0.383 | 0.418 | 0.402 | 0.414 | 0.429 | 0.439 | 0.512 | 0.493 | 0.400 | 0.435 |
| | 336 | 0.372 | 0.418 | 0.369 | 0.408 | 0.373 | 0.407 | 0.329 | 0.380 | 0.448 | 0.465 | 0.452 | 0.452 | 0.496 | 0.487 | 0.552 | 0.551 | 0.747 | 0.599 |
| | 720 | 0.407 | 0.447 | 0.403 | 0.439 | 0.406 | 0.441 | 0.379 | 0.422 | 0.605 | 0.551 | 0.462 | 0.468 | 0.463 | 0.474 | 0.562 | 0.560 | 1.454 | 0.847 |
| ETTm1 | 96 | 0.315 | 0.367 | 0.293 | 0.349 | 0.292 | 0.346 | 0.290 | 0.342 | 0.299 | 0.343 | 0.338 | 0.375 | 0.379 | 0.419 | 0.386 | 0.398 | 0.318 | 0.367 |
| | 192 | 0.352 | 0.387 | 0.326 | 0.368 | 0.332 | 0.372 | 0.332 | 0.369 | 0.335 | 0.365 | 0.374 | 0.387 | 0.426 | 0.441 | 0.459 | 0.444 | 0.355 | 0.391 |
| | 336 | 0.381 | 0.404 | 0.352 | 0.384 | 0.366 | 0.394 | 0.366 | 0.392 | 0.369 | 0.386 | 0.410 | 0.411 | 0.445 | 0.459 | 0.495 | 0.464 | 0.401 | 0.419 |
| | 720 | 0.437 | 0.436 | 0.405 | 0.416 | 0.417 | 0.421 | 0.416 | 0.420 | 0.425 | 0.421 | 0.478 | 0.450 | 0.543 | 0.490 | 0.585 | 0.516 | 0.448 | 0.448 |
| ETTm2 | 96 | 0.189 | 0.277 | 0.170 | 0.260 | 0.173 | 0.262 | 0.165 | 0.255 | 0.167 | 0.269 | 0.187 | 0.267 | 0.203 | 0.287 | 0.192 | 0.274 | 0.197 | 0.271 |
| | 192 | 0.252 | 0.318 | 0.227 | 0.297 | 0.229 | 0.301 | 0.220 | 0.292 | 0.224 | 0.303 | 0.249 | 0.309 | 0.269 | 0.328 | 0.280 | 0.339 | 0.285 | 0.328 |
| | 336 | 0.301 | 0.351 | 0.275 | 0.328 | 0.286 | 0.341 | 0.274 | 0.329 | 0.281 | 0.342 | 0.321 | 0.351 | 0.325 | 0.366 | 0.334 | 0.361 | 0.338 | 0.366 |
| | 720 | 0.382 | 0.405 | 0.363 | 0.387 | 0.378 | 0.401 | 0.362 | 0.385 | 0.397 | 0.421 | 0.408 | 0.403 | 0.421 | 0.415 | 0.417 | 0.413 | 0.395 | 0.419 |
| ILI | 24 | 2.822 | 1.142 | 2.728 | 1.114 | 2.063 | 0.881 | 1.319 | 0.754 | 2.215 | 1.081 | 2.317 | 0.934 | 3.228 | 1.260 | 2.294 | 0.945 | 4.539 | 1.528 |
| | 36 | 2.862 | 1.143 | 2.669 | 1.092 | 1.868 | 0.892 | 1.430 | 0.834 | 1.963 | 0.963 | 1.972 | 0.920 | 2.679 | 1.080 | 1.825 | 0.848 | 4.628 | 1.534 |
| | 48 | 2.846 | 1.123 | 2.728 | 1.098 | 1.790 | 0.884 | 1.553 | 0.815 | 2.130 | 1.024 | 2.238 | 0.940 | 2.622 | 1.078 | 2.010 | 0.900 | 4.957 | 1.585 |
| | 60 | 3.023 | 1.155 | 2.883 | 1.126 | 1.979 | 0.957 | 1.470 | 0.788 | 2.368 | 1.096 | 2.027 | 0.928 | 2.857 | 1.157 | 2.178 | 0.963 | 5.429 | 1.661 |

*Table 20.* Anomaly detection performance across 41 datasets from the UCR Anomaly Archive measured using adjusted best $F_1$ and VUS-ROC. Bold indicates the best performance per dataset/metric. Baselines from (Goswami et al., 2024).

| | Adjusted Best $F_1$ | | | | | VUS-ROC | | | | |
|---|---|---|---|---|---|---|---|---|---|---|
| | TiViT | Anomaly TF | Moment | GPT4TS | TimesNet | TiViT | AnomalyTF | Moment | GPT4TS | TimesNet |
| 1sddb40 | **0.935** | 0.030 | 0.540 | 0.190 | 0.680 | **0.772** | 0.640 | 0.750 | 0.660 | 0.720 |
| BIDMC1 | **1.000** | 0.990 | **1.000** | **1.000** | **1.000** | 0.642 | 0.690 | 0.650 | 0.630 | **0.740** |
| CHARISfive | 0.046 | 0.010 | **0.130** | 0.020 | 0.080 | **0.572** | 0.360 | 0.400 | 0.450 | 0.460 |
| CHARISten | **0.851** | 0.020 | 0.110 | 0.100 | 0.030 | **0.597** | 0.430 | 0.540 | 0.510 | 0.530 |
| CIMIS44AirTemperature3 | **1.000** | 0.060 | 0.980 | 0.180 | 0.470 | **0.843** | 0.640 | 0.750 | 0.620 | 0.740 |
| CIMIS44AirTemperature5 | **1.000** | 0.390 | 0.990 | 0.200 | 0.710 | **0.859** | 0.780 | 0.810 | 0.560 | 0.720 |
| ECG2 | **1.000** | **1.000** | **1.000** | 0.900 | **1.000** | 0.821 | 0.830 | **0.840** | 0.780 | 0.600 |
| ECG3 | **1.000** | 0.360 | 0.980 | 0.840 | 0.480 | **0.808** | 0.540 | 0.770 | 0.450 | 0.610 |
| Fantasia | **1.000** | 0.750 | 0.950 | 0.870 | 0.550 | **0.786** | 0.730 | 0.640 | 0.650 | 0.610 |
| GP711MarkerLFM5z4 | **1.000** | 0.930 | **1.000** | 0.640 | 0.950 | **0.886** | 0.540 | 0.730 | 0.620 | 0.720 |
| GP711MarkerLFM5z5 | **1.000** | 0.760 | 0.970 | 0.480 | 0.900 | **0.961** | 0.690 | 0.720 | 0.630 | 0.840 |
| InternalBleeding5 | **1.000** | 0.940 | **1.000** | 0.920 | **1.000** | 0.932 | 0.460 | 0.690 | 0.630 | **0.940** |
| Italianpowerdemand | 0.310 | 0.010 | **0.740** | 0.010 | 0.440 | 0.709 | 0.450 | **0.770** | 0.480 | 0.710 |
| Lab2Cmac011215EPG5 | **1.000** | 0.990 | 0.980 | 0.600 | 0.990 | 0.739 | **0.770** | 0.630 | 0.640 | 0.610 |
| Lab2Cmac011215EPG6 | 0.267 | **0.410** | 0.100 | 0.100 | 0.170 | 0.554 | **0.700** | 0.480 | 0.520 | 0.450 |
| MesoplodonDensirostris | **1.000** | **1.000** | 0.840 | **1.000** | **1.000** | 0.748 | **0.850** | 0.720 | 0.690 | 0.790 |
| PowerDemand1 | **0.994** | 0.870 | 0.440 | 0.760 | 0.950 | **0.919** | 0.720 | 0.540 | 0.600 | 0.750 |
| TkeepFirstMARS | **0.577** | 0.010 | 0.150 | 0.020 | 0.230 | 0.728 | 0.520 | 0.760 | 0.500 | **0.790** |
| TkeepSecondMARS | **1.000** | 0.830 | **1.000** | 0.120 | 0.950 | **0.989** | 0.720 | 0.910 | 0.810 | 0.980 |
| WalkingAceleration5 | 0.967 | 0.990 | **1.000** | 0.870 | 0.930 | **0.968** | 0.940 | 0.870 | 0.910 | 0.850 |
| apneaecg | **0.814** | 0.400 | 0.200 | 0.310 | 0.260 | 0.608 | 0.580 | 0.690 | 0.580 | **0.760** |
| apneaecg2 | **1.000** | 0.650 | **1.000** | **1.000** | 0.650 | **0.845** | 0.790 | 0.740 | 0.650 | 0.610 |
| gait1 | **1.000** | 0.180 | 0.360 | 0.410 | 0.520 | **0.887** | 0.630 | 0.570 | 0.580 | 0.600 |
| gaitHunt1 | **0.596** | 0.080 | 0.430 | 0.100 | 0.300 | **0.847** | 0.810 | 0.680 | 0.710 | 0.840 |
| insectEPG2 | **0.962** | 0.120 | 0.230 | 0.810 | 0.960 | **0.871** | 0.650 | 0.820 | 0.560 | 0.730 |
| insectEPG4 | 0.513 | 0.980 | **1.000** | 0.210 | 0.850 | 0.691 | 0.690 | **0.720** | 0.490 | 0.650 |
| ltstdbs30791AS | **1.000** | **1.000** | **1.000** | **1.000** | **1.000** | **0.959** | 0.780 | 0.810 | 0.740 | 0.670 |
| mit14046longtermecg | **0.676** | 0.450 | 0.590 | 0.580 | 0.600 | 0.661 | 0.790 | 0.660 | 0.610 | **0.840** |
| park3m | **1.000** | 0.150 | 0.640 | 0.630 | 0.930 | **0.875** | 0.630 | 0.780 | 0.540 | 0.780 |
| qtdbSel1005V | **0.844** | 0.410 | 0.650 | 0.390 | 0.530 | 0.612 | 0.520 | **0.640** | 0.610 | 0.540 |
| qtdbSel100MLII | **1.000** | 0.420 | 0.840 | 0.600 | 0.870 | 0.573 | 0.620 | 0.620 | 0.580 | **0.650** |
| resperation1 | 0.308 | 0.000 | 0.150 | 0.010 | 0.030 | 0.725 | **0.750** | 0.670 | 0.470 | 0.670 |
| s20101mML2 | **1.000** | 0.690 | 0.710 | 0.050 | 0.080 | **0.942** | 0.640 | 0.720 | 0.640 | 0.690 |
| sddb49 | **1.000** | 0.890 | **1.000** | 0.940 | **1.000** | **0.937** | 0.660 | 0.730 | 0.580 | 0.680 |
| sel840mECG1 | **0.984** | 0.160 | 0.660 | 0.210 | 0.360 | 0.702 | 0.620 | **0.720** | 0.650 | 0.600 |
| sel840mECG2 | **0.984** | 0.150 | 0.390 | 0.280 | 0.210 | 0.683 | 0.590 | **0.690** | 0.520 | 0.520 |
| tilt12744mtable | **0.254** | 0.070 | 0.240 | 0.000 | 0.030 | **0.761** | 0.480 | 0.740 | 0.510 | 0.640 |
| tilt12754table | 0.131 | 0.230 | **0.640** | 0.060 | 0.050 | **0.855** | 0.600 | 0.820 | 0.550 | 0.750 |
| tiltAPB2 | **1.000** | 0.920 | 0.980 | 0.830 | 0.380 | **0.844** | 0.770 | 0.770 | 0.600 | 0.700 |
| tiltAPB3 | 0.148 | 0.170 | **0.850** | 0.050 | 0.090 | **0.769** | 0.680 | 0.650 | 0.440 | 0.580 |
| weallwalk | **0.706** | 0.000 | 0.580 | 0.130 | 0.170 | 0.849 | 0.730 | **0.930** | 0.870 | 0.850 |
| Mean | **0.802** | 0.475 | 0.684 | 0.449 | 0.570 | **0.789** | 0.659 | 0.711 | 0.605 | 0.695 |
| Median | **0.994** | 0.410 | 0.740 | 0.390 | 0.550 | **0.808** | 0.660 | 0.720 | 0.600 | 0.700 |
| Std | 0.300 | 0.379 | 0.321 | 0.358 | 0.355 | 0.122 | 0.124 | 0.106 | 0.107 | 0.118 |

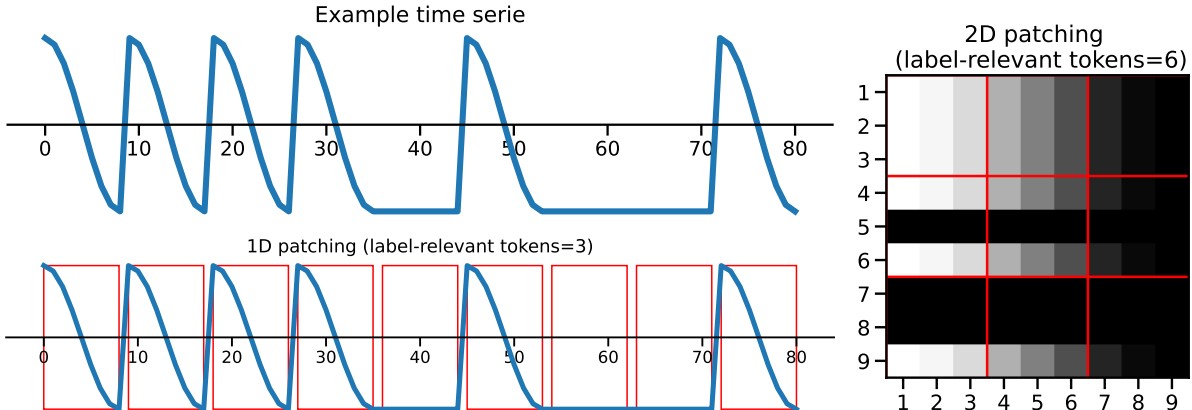

*Figure 13.* 2D patching yields a higher number of label-relevant tokens (with constant negative signal) than 1D patching. This facilitates time series classification with Transformers.

## D. Insights on Modeling Time Series as Images

TiViT surpasses the performance of TSFMs in time series tasks by leveraging pretrained ViTs. This raises a key question: is its success solely due to the rich representations learned from billions of natural images, or is there an inherent advantage of the 2D patching strategy as well? We develop a theoretical insight at the patch level showing how the 2D representation of time series can enhance the classification performance of Transformer models. To empirically validate this, we compare the performance of Transformers pretrained on real-world data using 1D versus 2D patching.

### D.1. Theoretical Analysis of 1D and 2D Patching

We consider a binary time series classification problem with $N$ univariate training samples $\{(\boldsymbol{t}^n, y^n), y^n \in \{+1, -1\}\}_{n=1}^N$. Each time series $\boldsymbol{t}^n \in \mathbb{R}^T$ can be patched as follows:

- 1D patching: The series $\boldsymbol{t}$ is split into $k$ contiguous, non-overlapping tokens $\boldsymbol{x}_l \in \mathbb{R}^k$.

- 2D patching: The series $\boldsymbol{t}$ is reshaped into a $k \times k$ matrix, then divided into $k$ non-overlapping $\sqrt{k} \times \sqrt{k}$ patches, which are flattened to form tokens $\boldsymbol{x}'_{(i,j)} \in \mathbb{R}^k$.

This setup ensures the same number of tokens for 1D and 2D patching. Our analysis builds on the notion of label-relevant tokens introduced by Li et al. (2023a) (see Section D.2). Following their data model, we consider each token to be a noisy version of distinct patterns. In binary classification, there exist two such patterns $\{\boldsymbol{\mu}_1, \boldsymbol{\mu}_2\}$, $\boldsymbol{\mu}_i \in \mathbb{R}^k, \forall\, i$. For a time series $\boldsymbol{t}^n$ with label $y^n = 1$, tokens $\boldsymbol{x}$ that are noisy $\boldsymbol{\mu}_1$, i.e., $\|\boldsymbol{x} - \boldsymbol{\mu_1}\| \leq \|\boldsymbol{x} - \boldsymbol{\mu_2}\|$, are label-relevant. Similarly, for a time series $\boldsymbol{t}^n$ with label $y^n = -1$, the noisy versions of $\boldsymbol{\mu}_2$ are label-relevant.

**Benefits of 2D patching.** Li et al. (2023a) showed that the sample complexity of a Transformer scales as $\mathcal{O}(1/\alpha_*^2)$ where $\alpha_*$ denotes the fraction of label-relevant tokens in the training samples. In Section D.3, we provide a constructive proof showing that under certain conditions, this fraction of label-relevant tokens is greater when the time series is transformed into a 2D representation compared to the conventional 1D representation. Therefore, 2D patching can lead to more efficient learning with Transformers than 1D patching. Figure 13 illustrates our idea for an exemplary time series with $T = 91$ and $k = 9$. We set $\boldsymbol{\mu}_1 = \cos(x)$ for $x \in [0, \pi]$ and define the label-relevant signal as $\boldsymbol{\mu}_2 = -1$. In the 1D case, only three tokens carry the label-relevant information, whereas in the 2D case there are six such tokens. Following Li et al. (2023a), distributing the discriminative signal across a larger number of tokens makes it easier for a Transformer to detect and leverage it.

Below, we review the shallow ViT and data model introduced by Li et al. (2023a) in their theoretical analysis of training a ViT. Their Theorem D.2 shows that the sample complexity for ViTs to achieve a zero generalization error is inversely correlated with the fraction of label-relevant tokens. Building on this insight, we introduce and prove Proposition 1, showing that 2D patching can increase the number of label-relevant tokens compared to 1D patching.

*Table 21.* Key Notations

| Notation | Description |
| --- | --- |
| $\alpha_*$ | Fraction of label-relevant tokens |
| $\sigma, \delta, \tau$ | Initialization/token noise parameters |
| $\kappa$ | Minimum pattern distance |
| $M$ | Total number of patterns |

## D.2. Theoretical Background

**Model and setup.** Following the setup of Li et al. (2023a), we study a binary classification problem with $N$ training samples $\{(\boldsymbol{X}^n, y^n)\}_{n=1}^N$. Each input $\boldsymbol{X}^n \in \mathbb{R}^{d \times L}$ contains $L$ tokens $\{\boldsymbol{x}_1^n, \ldots, \boldsymbol{x}_L^n\}$. Labels $y^n \in \{\pm 1\}$ are determined by majority vote over discriminative tokens. A simplified Vision Transformer (ViT) (Dosovitskiy et al., 2021) model is defined as:

$$F(\boldsymbol{X}^n) = \frac{1}{|\mathcal{S}^n|} \sum_{l \in \mathcal{S}^n} \boldsymbol{a}_{(l)}^\top \mathrm{ReLU}\left(\boldsymbol{W}_O \boldsymbol{W}_V \boldsymbol{X}^n \mathrm{softmax}\left(\boldsymbol{X}^{n\top} \boldsymbol{W}_K^\top \boldsymbol{W}_Q \boldsymbol{x}_l^n\right)\right),$$

where $\psi = (\boldsymbol{A} = \{\boldsymbol{a}_{(l)}\}_l, \boldsymbol{W}_O, \boldsymbol{W}_V, \boldsymbol{W}_K, \boldsymbol{W}_Q)$ are trainable parameters. The empirical risk minimization problem is:

$$\min_\psi f_N(\psi) = \frac{1}{N} \sum_{n=1}^N \max\left\{1 - y^n \cdot F(\boldsymbol{X}^n), 0\right\}.$$

Training uses mini-batch SGD with fixed output layer weights $\boldsymbol{A}$, following standard NTK initialization practices.

**Data model.** Tokens $\boldsymbol{x}_l^n$ are noisy versions of $M$ patterns $\{\boldsymbol{\mu}_1, \ldots, \boldsymbol{\mu}_M\}$, where $\boldsymbol{\mu}_1, \boldsymbol{\mu}_2$ are discriminative. Label $y^n$ depends on majority vote over tokens closest to $\boldsymbol{\mu}_1/\boldsymbol{\mu}_2$. Noise level $\tau$ satisfies $\tau < \kappa/4$, with $\kappa - 4\tau = \Theta(1)$.

**Generalization of ViT.** We now recap the main results from Li et al. (2023a) from which we derive our result, along with the main notations in Table 21.

**Assumption** (Initial Model Conditions, (Li et al., 2023a)). Initial weights $\boldsymbol{W}_V^{(0)}, \boldsymbol{W}_K^{(0)}, \boldsymbol{W}_Q^{(0)}$ satisfy:

$$\|\boldsymbol{W}_V^{(0)} \boldsymbol{\mu}_j - \boldsymbol{p}_j\| \le \sigma, \quad \|\boldsymbol{W}_K^{(0)} \boldsymbol{\mu}_j - \boldsymbol{q}_j\| \le \delta, \quad \|\boldsymbol{W}_Q^{(0)} \boldsymbol{\mu}_j - \boldsymbol{r}_j\| \le \delta,$$

for orthonormal bases $\mathcal{P}, \mathcal{Q}, \mathcal{R}$ and $\sigma = O(1/M), \delta < 1/2$.

**Theorem** (Generalization of ViT, (Li et al., 2023a)). Under Assumption 1, with sufficient model width $m \gtrsim \epsilon^{-2} M^2 \log N$, fraction

$$\alpha_* \ge \alpha_\# / (\epsilon_S e^{-(\delta+\tau)} (1 - (\sigma + \tau)),$$

and sample size

$$N \ge \Omega\left((\alpha_* - c'(1 - \zeta) - c''(\sigma + \tau))^{-2}\right),$$

SGD achieves zero generalization error after

$$T = \Theta\left(\frac{1}{(1 - \epsilon - (\sigma + \tau) M/\pi) \eta \alpha_*}\right)$$

iterations.

**Proposition** (Generalization without Self-Attention, (Li et al., 2023a)). Without self-attention, achieving zero error requires $N \ge \Omega\left((\alpha_*(\alpha_* - \sigma - \tau))^{-2}\right)$, demonstrating ViT's sample complexity reduction by $1/\alpha_*^2$.

## D.3. Proof of Label Relevance in 2D Patches

We introduce Proposition 1 that formalizes our theoretical analysis of 1D and 2D patching from Section D.1 and provide a detailed proof.

**Proposition 1.** *For an arbitrary $\boldsymbol{\mu}_1, \boldsymbol{\mu}_2 \in \mathbb{R}^k$, let $\boldsymbol{t} = [\boldsymbol{x}_1 \ \boldsymbol{x}_2 \ \cdots \ \boldsymbol{x}_k]^\top \in \mathbb{R}^T$where $\forall i \in [k], \boldsymbol{x}_i \in \mathbb{R}^k$ and either $\boldsymbol{x}_i = \boldsymbol{\mu}_1$ or $\boldsymbol{x}_i = \boldsymbol{\mu}_2$ with $\boldsymbol{\mu}_2$ being a label-relevant pattern. Let $|\{i : \boldsymbol{x}_i = \boldsymbol{\mu}_2\}| = n'$ and assume that $2\boldsymbol{x}' \cdot (\boldsymbol{\mu}_1 - \boldsymbol{\mu}_2) \leq ||\boldsymbol{\mu}_1||^2 - ||\boldsymbol{\mu}_2||^2$ whenever $|\{i : x_i' \in \boldsymbol{\mu}_2\}| \geq \sqrt{k}$. Then, it holds that*

$$\alpha_*^{2D} \geq \alpha_*^{1D} = \frac{n'}{k},$$

*and the inequality is strict if $n' \bmod \sqrt{k} > 0$.*

*Proof.* For a token $\boldsymbol{x}'^n$ to be label-relevant (aligned with $\boldsymbol{\mu}_2$), it must satisfy:

$$\|\boldsymbol{x}'^n - \boldsymbol{\mu}_2\| \leq \|\boldsymbol{x}'^n - \boldsymbol{\mu}_1\|.$$

Expanding both sides, we have that:

$$\|\boldsymbol{x}'^n\|^2 + 2\boldsymbol{x}'^n \cdot \boldsymbol{\mu}_1 + \|\boldsymbol{\mu}_1\|^2 \leq \|\boldsymbol{x}'^n\|^2 - 2\boldsymbol{x}'^n \cdot \boldsymbol{\mu}_2 + \|\boldsymbol{\mu}_2\|^2.$$

Regrouping the terms gives us the desired condition:

$$2\boldsymbol{x}'^n \cdot (\boldsymbol{\mu}_1 - \boldsymbol{\mu}_2) \leq ||\boldsymbol{\mu}_1||^2 - ||\boldsymbol{\mu}_2||^2. \tag{1}$$

Recall that $n'$ denotes the number of segments of $\boldsymbol{\mu}_2$ in time series $\boldsymbol{t}$. Each such segment spans $\sqrt{k}$ tokens, contributing at least $\sqrt{k}$ elements to each of them. Under the assumption of the proposition, it implies (1) and makes each of these $\sqrt{k}$ tokens label-relevant.

We now need to carefully consider how the $\boldsymbol{\mu}_2$ segments can be placed within $\boldsymbol{t}$ to understand how many tokens become label-relevant thanks to each $\boldsymbol{\mu}_2$. We consider two cases: 1) $n' = c\sqrt{k}$ for some $c \in \mathbb{N}$ satisfying $n' \in (0, k]$, and 2) $n' = c\sqrt{k} + b$ for some $a, b \in \mathbb{N}, \sqrt{k} > b > 0$ such that $n' \in (0, k]$. In the first case, $\alpha_*^{1D} = c\sqrt{k}/k$. In the case of 2D patching, in the worst case, $\boldsymbol{\mu}_2$ segments can be placed such that they will contribute to $c\sqrt{k}$ tokens. In this case, $\alpha_*^{2D} \geq c\sqrt{k}/k$ and $\alpha_*^{1D} \leq \alpha_*^{2D}$. If $n'$ is not a multiple of $\sqrt{k}$, the same analysis applies for the $c\sqrt{k}$ segments of $\boldsymbol{\mu}_2$. To account for the remainder $b$, we note that for any $b > 0$, in 2D case, it adds $\sqrt{k}$ label-relevant tokens to the fraction $\alpha_*^{2D}$ so that $\alpha_*^{2D} \geq \frac{c\sqrt{k}+\sqrt{k}}{k}$. In the case of 1D patching, $\alpha_*^{1D} = \frac{c\sqrt{k}+b}{k}$. Given that $b < \sqrt{k}$, this concludes the proof. $\square$

To illustrate the benefits of 2D modeling and patching, we present several examples of time series in Figure 15. We define $\boldsymbol{\mu}_1$ using functions such as log, cosine, and sine. We then set $\boldsymbol{\mu}_2 = \mathbf{1}_k$, $n' = 3$ and randomly shuffle $\boldsymbol{\mu}_1$ and $\boldsymbol{\mu}_2$ segments within the generated input time series.

### D.4. Interpretability Scores

We illustrate the spread of information achieved with 2D modeling on samples from a real-world dataset from the UCR repository. In particular, we show that a model trained on 2D representations of time series has more regions that it deems relevant for predicting the class membership of the time series. To this end, we follow (Early et al., 2024) and use MILLET: a framework that provides interpretability scores for timestamps within a time series given a pretrained model. For this, we train two shallow ViTs, ViT1D and ViT2D, on the BirdChicken dataset from the UCR repository. ViT1D takes as input a raw 1D time series, while ViT2D is trained on square 2D images of the time series. The only difference between these two models is their patching strategy: ViT1D patches the time series using a 1D convolutional filter, while ViT2D applies a 2D convolutional filter. The obtained results for two samples from different classes are presented in Figure 14, for 1D (left) and 2D (right) cases, respectively. Note that the 1D heatmap for the sample with ground-truth class 0 highlights discriminative signals for class 1 at the beginning and end of the time series, while the corresponding 2D heatmap displays no such signals in these areas. The interpretability scores w.r.t. the ground-truth class of a sample are generally more homogeneous for ViT2D which facilitates classification.

### D.5. Pretraining Transformers with 1D and 2D Patching

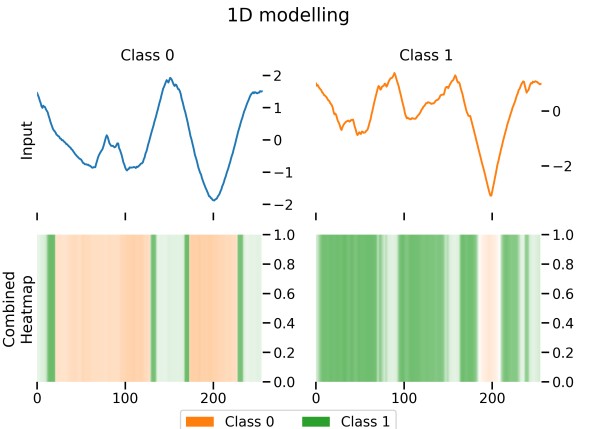
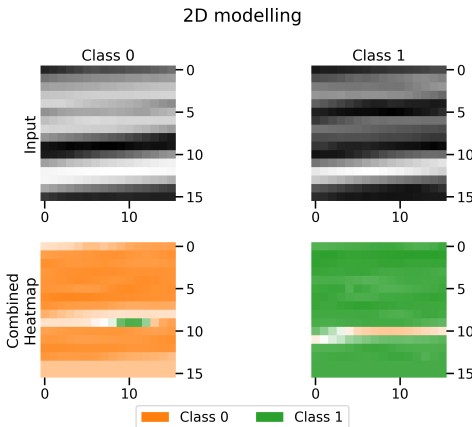

*Figure 14.* Comparison of interpretability heatmaps for two ViTs trained on 1D (left) and 2D (right) representations of time series from the BirdChicken dataset, respectively. The interpretability scores for the correct class of a sample are more homogeneous in the 2D case.

To validate our hypothesis that 2D patching improves representation learning for time series, we study the effect of different patching strategies while keeping the Transformer architecture and pretraining objective fixed.

**Model.** We adopt the setup of Feofanov et al. (2025), whose Transformer block implementation is closely related to the classical ViT (ViTUnit class here). Specifically, the model consists of 6 Transformer layers, each with 8 attention heads and an embedding dimension of 256. We pretrain all models with contrastive learning following Feofanov et al. (2025) and He et al. (2020). The augmentation technique to generate positive pairs is RandomCropResize with a crop rate varying within $[0, 0.2]$. All time series are resized to a fixed length $T = 512$ using interpolation.

**1D and 2D patching.** We compare 1D and 2D patching in both non-overlapping and overlapping settings. For non-overlapping 1D patching, we generate 32 patches of size 16. For non-overlapping 2D patching, we first arrange the 1D patches in a matrix of size $32 \times 16$ and then extract 32 patches of size $2 \times 8$. After flattening, we obtain 32 patches of size 16, similar to the 1D setting, but semantically different. For overlapping 1D patching, we apply a stride of 8, which yields 64 patches of size 16. For overlapping 2D patching, we rearrange these 1D patches again in a matrix of size $64 \times 16$ and then extract 32 patches of size $4 \times 8$. Flattening yields 32 patches of size 32.

**Data.** For pretraining, we construct a corpus of 100,000 samples from publicly available time series datasets that do not overlap with the evaluation benchmark. Specifically, the corpus is assembled from ECG (Clifford et al., 2017), EMG (Goldberger et al., 2000), Epilepsy (Andrzejak et al., 2001), FD-A and FD-B (Lessmeier et al., 2016), Gesture (Liu et al., 2009), HAR (Anguita et al., 2013), and SleepEEG (Kemp et al., 2000). To reduce computational cost, we use a balanced subset of the full collection. Detailed sample counts per source dataset are reported in Table 23.

**Classification results.** We evaluate the learned representations on the UCR benchmark. As shown in Table 22, 2D patching consistently outperforms 1D patching, with overlapping 2D patches achieving the highest classification accuracy. These results indicate that transforming time series into image-like structures is beneficial not only when leveraging pretrained ViTs, but also when pretraining Transformers from scratch on time series data.

*Table 22.* Evaluation of models pretrained with different patching strategies on UCR.

| Patching | Non-overlap | | Overlap | |
|---|---|---|---|---|
| | 1D | 2D | 1D | 2D |
| Accuracy | 76.4 | 76.8 | 76.6 | **77.4** |

# E. Broader Impact

Since this paper presents foundational machine learning research, we do not see any direct societal risks. The broader impact of our work will depend on its specific application.

We demonstrate that our method TiViT significantly improves classification accuracy. This advancement can be beneficial in healthcare where the analysis of physiological signals is crucial for early diagnosis and treatment or in industry where the

*Table 23.* Data used to pretrain Transformers for comparison of 1D and 2D patching.

| Dataset | Number of examples | Prop. of taken examples |
|---------|--------------------|-----------------------| 
| ECG | 20835 | 45.7% |
| EMG | 163 | 100% |
| Epilepsy | 11480 | 100% |
| FD-A | 10912 | 100% |
| FD-B | 13619 | 100% |
| Gesture | 1320 | 100% |
| HAR | 20835 | 78.7% |
| SleepEEG | 20836 | 4.5% |

accurate monitoring of sensor data enables predictive maintenance and reduces downtime.

However, deep learning models including TiViT operate as black boxes with limited interpretability. In safety-critical domains or applications directly impacting humans, such models necessitate careful deployment and oversight. Further research into interpretability and human-in-the-loop frameworks is essential to make deep learning models trustworthy for real-world settings.

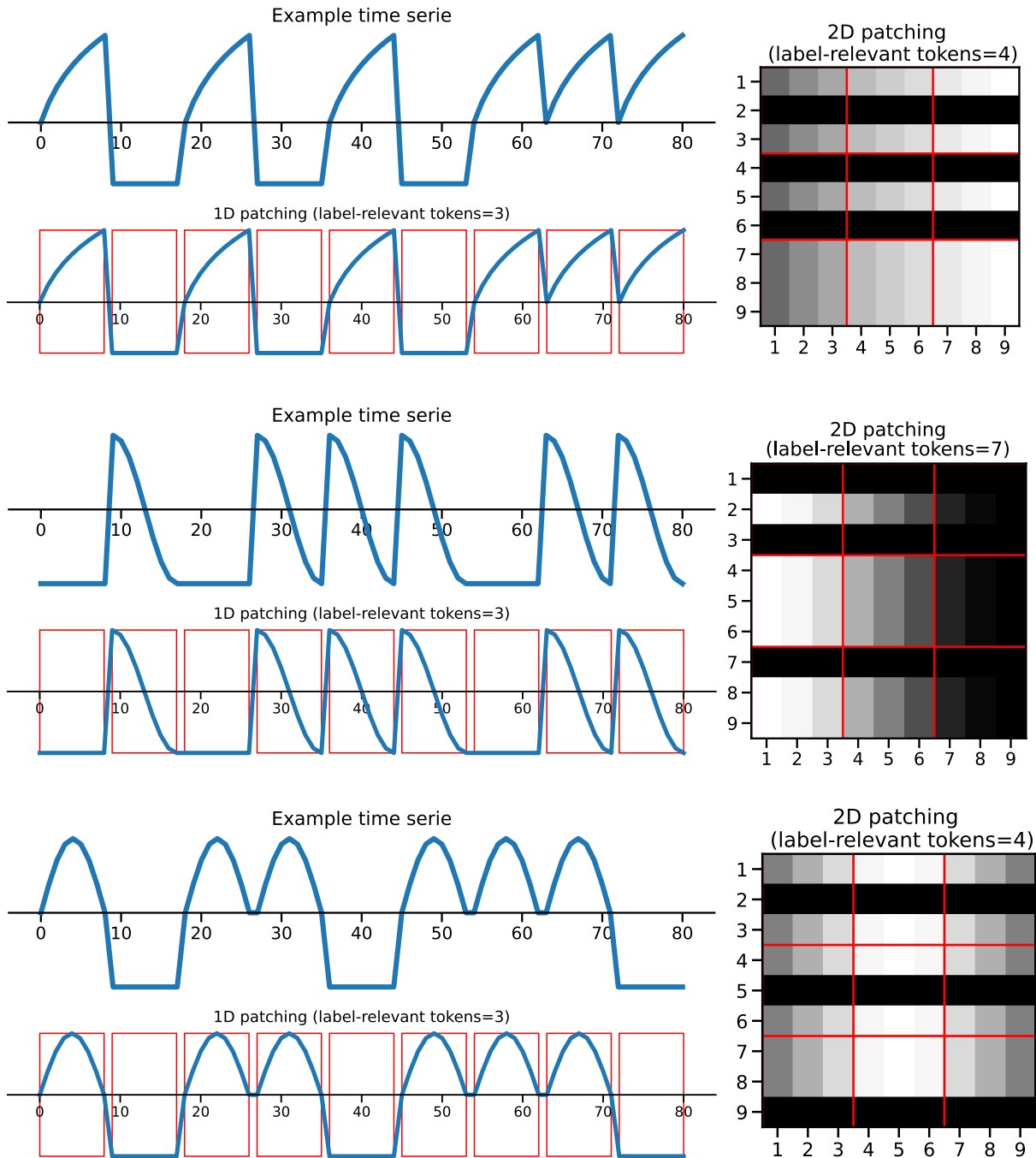

*Figure 15.* Illustration of Proposition 1 on more generated time series. In each example considered, 2D patching is more beneficial due to the higher number of label-relevant tokens.

