# OpenReview forum: "TiViT: Time Series Representations for Classification Lie Hidden in Pretrained Vision Transformers"
_ICML.cc/2026/Workshop/FMSD — FMSD @ ICML 2026 Poster_

### Official Review · Reviewer_6qef · 2026-05-20
**TiViT: Time Series Representations for Classification Lie Hidden in Pretrained Vision Transformers**

**Rating:** 7
**Confidence:** 4

**Review:**

### **Summary:**
- TiViT shows that frozen pretrained ViT hidden representations can be effectively reused for time-series classification, often outperforming existing TSFMs and complementing their representations.

### **Strengths:**
- TiViT achieves strong classification performance on UCR, UEA and further improves when combined with TSFM representations.
- Representation alignment study provides evidence that TiViT and TSFMs capture complementary information.
- Continual pretraining experiment shows that ViT backbones can be adapted to time series with synthetic data and LoRA-based training.

### **Weaknesses:**
- TiViT relies on large vision-pretrained backbones, while the TSFM baselines have relatively smaller parameter scales and are trained with smaller pretraining corpora.
- TiViT method depends on several design choices, including image transformation, patch size, overlap, interpolation, and hidden-layer selection.

---

### Official Review · Reviewer_GtMs · 2026-05-21
**TiViT: Time Series Representations for Classification Lie Hidden in Pretrained Vision Transformers**

**Rating:** 5
**Confidence:** 4

**Review:**

## Summary

The paper proposes TiViT, which converts time series into 2D grayscale images by stacking overlapping patches and extracts features from intermediate layers of a frozen pretrained ViT (OpenCLIP, SigLIP 2, DINOv3) with a linear classifier on top.

## Strengths

- Clear writing with well-structured ablations across patch size, overlap, interpolation, imaging method, aggregation, and backbone size.
- Intrinsic dimension analysis (ρ=0.704 correlation with accuracy) provides a non-trivial mechanistic explanation for why intermediate layers work best.


## Areas for Improvement

- The "first" framing is overclaimed : several prior works already apply pretrained vision models to time series, including [1, 2, 3]. The broader idea also extends to VLMs [4, 5] beyond ViTs. The claim should be scoped to "first to use frozen hidden-layer ViT representations for**time-series classification**".
- Use the UCR/UEA bake-off baselines for a realistic comparison : HIVE-COTE 2.0, ROCKET, MultiROCKET, InceptionTime etc. are the standard reference points. Given HIVE-COTE 2.0 reaches ~89% on UCR vs TiViT's 81.6%, the "state-of-the-art" claim either needs to be scoped to "among foundation-model approaches" or substantiated against current bake-off SOTA.




**References**


[1] Chen et al. 2024. VisionTS: Visual Masked Autoencoders Are Free-Lunch Zero-Shot Time Series Forecasters.

[2] Li et al. 2023. Time Series as Images: Vision Transformer for Irregularly Sampled Time Series. NeurIPS 2023.

[3] Prithyani et al. 2024. On the Feasibility of Vision-Language Models for Time-Series Classification.

[4] Liu et al. 2025. MLLM4TS: Leveraging Vision and Multimodal Language Models for General Time-Series Analysis.

[5] Ni et al. 2025. Harnessing Vision Models for Time Series Analysis: A Survey.

---

### Official Review · Reviewer_TAe5 · 2026-05-22
**A simple and practically effective framework for time series classification**

**Rating:** 7
**Confidence:** 4

**Review:**

The paper proposes TiViT (Time Vision Transformer), a framework which applies the large pretrained vision transformers (ViTs) directly for time series. It shows good performance for time series classification. In the appendix, the framework is also tested for time series anomaly detection and forecasting. However, the performance of the proposed framework for forecasting is not as good as classification in comparison with other SOTA methods. Specifically, the time series are normalized and converted into images, which as feeded into frozen pretrained ViTs.

Pros:
1. A simple and practical useful framework for time series classification.
2. Strong empirical performance for time series classification.

Cons:
1. The empirical performance for time series forecasting is not outstanding.
2. The analysis for why the proposed framework is more suitable for time series classification than forecasting would make the paper more interesting.